# Adversarial Attacks on Multimodal Large Language Models: A Comprehensive Survey

**Bhavuk Jain**                                          *bhavukj@google.com*
*Google*

**Sercan Ö. Arık**                                       *soarik@google.com*
*Google*

**Hardeo K. Thakur**                                     *Hardeo.Thakur@bennett.edu.in*
*Bennett University, India*

**Reviewed on OpenReview:** *https://openreview.net/forum?id=zwzodDJkzZ*

## Abstract

Multimodal large language models (MLLMs) integrate information from multiple modalities such as text, images, audio, and video, enabling complex capabilities such as visual question answering and audio translation. While powerful, this increased expressiveness introduces new and amplified vulnerabilities to adversarial manipulation. This survey provides a comprehensive and systematic analysis of adversarial threats to MLLMs, moving beyond enumerating attack techniques to explain the underlying causes of model susceptibility. We introduce a taxonomy that organizes adversarial attacks according to attacker objectives, unifying diverse attack surfaces across modalities and deployment settings. Additionally, we also present a vulnerability-centric analysis that links integrity attacks, safety and jailbreak failures, control and instruction hijacking, and training-time poisoning to shared architectural and representational weaknesses in multimodal systems. Together, this framework provides an explanatory foundation for understanding adversarial behavior in MLLMs and informs the development of more robust and secure multimodal language systems.

## 1 Introduction

The rapid development of Large Language Models (LLMs) marks a significant leap in AI (Chang et al., 2023). By integrating and processing information from diverse modalities, such as text, images, and audio, in various combinations, these models mimic human perception to achieve a more holistic understanding of the world (Li et al., 2023). This advanced capability enables them to tackle complex tasks previously beyond reach, driving breakthroughs in areas such as image captioning (Vinyals et al., 2015), visual question answering (Antol et al., 2015), audio-visual scene analysis (Arandjelović & Zisserman, 2017), robotics (Mon-Williams et al., 2025), and more. The power of these models stems from their scalable attention-based architectures (Vaswani et al., 2017) and effective training recipes: pre-training on vast datasets, followed by post-training alignment on a multitude of tasks to suit downstream use cases, thereby achieving state-of-the-art performance. However, the very complexity that fuels the power of LLMs also introduces novel and intricate vulnerabilities (Goodfellow, 2014). As illustrated in Figure 1, unlike text-only LLMs, multimodal LLMs present an expanded attack surface where threats can arise not only from weaknesses within individual modality processing but, crucially, from the complex interplay and fusion mechanisms between the combined modalities (Baltrušaitis et al., 2018). These cross-modal vulnerabilities can manifest through several key attack vectors, some of which are as follows:

- **Cross-modal prompt injection** (Bagdasaryan et al., 2023) exploits LLMs' instruction-following nature by embedding malicious commands within non-textual modalities. Attackers can hide instructions in images or audio that the model interprets as textual commands, effectively hijacking behavior without directly manipulating text prompts.

- **Fusion mechanism attacks** target how the core process LLMs integrate information from multiple modalities, exploiting vulnerabilities in how features from different sources are combined and aligned. These attacks can disrupt the fusion process, causing the model to misinterpret or improperly weigh multimodal inputs.

- **Adversarial illusions** exploit the shared embedding space where LLMs align representations across modalities. Attackers can craft inputs in one modality that appear benign but whose embeddings become deceptively aligned with unrelated concepts from another modality, causing the model to hallucinate false semantic connections (Bagdasaryan et al., 2024).

As such powerful models are increasingly deployed in real-world applications, including safety-critical systems, understanding their susceptibility to adversarial manipulation becomes paramount.

The landscape of attacks targeting these LLMs is evolving rapidly, encompassing a range of techniques. These include adversarial perturbations designed to cause misclassification or erroneous outputs (Goodfellow et al., 2014), jailbreak attacks that bypass safety alignments (Wei et al., 2023), prompt injection methods that hijack model behavior (Perez & Ribeiro, 2022), and data poisoning strategies that corrupt the model during training (Gu et al., 2017). Given the increasing sophistication and potential impact of these attacks, a systematic understanding of the current threat landscape is urgently needed.

While recent efforts have begun to map the field with surveys on general LLM attacks (Shayegani et al., 2023b) and on specific pairs of modalities, such as vision–language models (Liu et al., 2024b), a deeper analytical connection between documented attacks and the underlying vulnerabilities they exploit remains limited. A notable peer-reviewed overview by Li & Fung (2025) surveys a broad range of security concerns for large language models, including prompt injection, adversarial manipulation, poisoning, and agent-related risks, providing a valuable high-level synthesis of the evolving threat landscape. More narrowly, recent work on multimodal prompt injection characterizes injection vectors and defenses across modality pairs, but does not pursue a unified taxonomy or vulnerability-level explanation beyond prompt-based control failures (Yeo & Choi, 2025). However, existing surveys predominantly organize attacks by modality, attack surface, or application context, and focus on cataloging attack techniques and defenses rather than systematically explaining why diverse attacks recur across different model architectures and deployment settings. To complement this line of work, our survey introduces a taxonomy that classifies adversarial attacks by attack objective, providing a unified organizational framework that cuts across modalities and application domains. Building on this taxonomy, we also adopt a vulnerability-focused perspective that explicitly links attack categories to shared architectural and representational weaknesses in multimodal large language models, including failures in cross-modal interaction, modality-specific processing, instruction following, and control. This combined taxonomy and vulnerability-driven analysis moves beyond enumerating attack instances to explain the root causes of adversarial susceptibility in MLLMs.

The remainder of this survey is organized as follows. Section 2 provides background on multimodal large language models, highlighting architectural components and formal abstractions that are most relevant to adversarial analysis. Section 3 introduces a goal-driven taxonomy of adversarial attacks on multimodal LLMs, organizing prior work by primary adversarial objective, and analyzes these attacks under different attacker knowledge assumptions (white-box, gray-box, and black-box). Section 4 shifts the focus from attack constructions to root causes, presenting a vulnerability-centric analysis that systematically links observed attack families to shared architectural, representational, and training-induced weaknesses in multimodal systems. Section 5 briefly discusses representative defense mechanisms and mitigation strategies, contextualized with respect to the attack families in the proposed taxonomy. Section 6 outlines the limitations of this survey and Section 7 discusses the broader impact of this survey paper. The Appendix in the end, details the survey methodology and inclusion criteria, and provides a consolidated table summarizing empirical characteristics of the works covered in the taxonomy diagram.

In terms of paper selection, this survey focuses on peer-reviewed works that evaluate adversarial attacks on multimodal large language models with a language-model-based reasoning core processing two or more input modalities. Attacks operating on unimodal surfaces are included when they are evaluated against an MLLM, as they represent baseline threats inherited by multimodal systems. Conversely, works targeting unimodal-only models, non-LLM multimodal systems, or non-adversarial failure modes (e.g., calibration, fairness) are excluded, as are non-peer-reviewed preprints and non-English publications. Full inclusion and exclusion criteria are detailed in Appendix A.

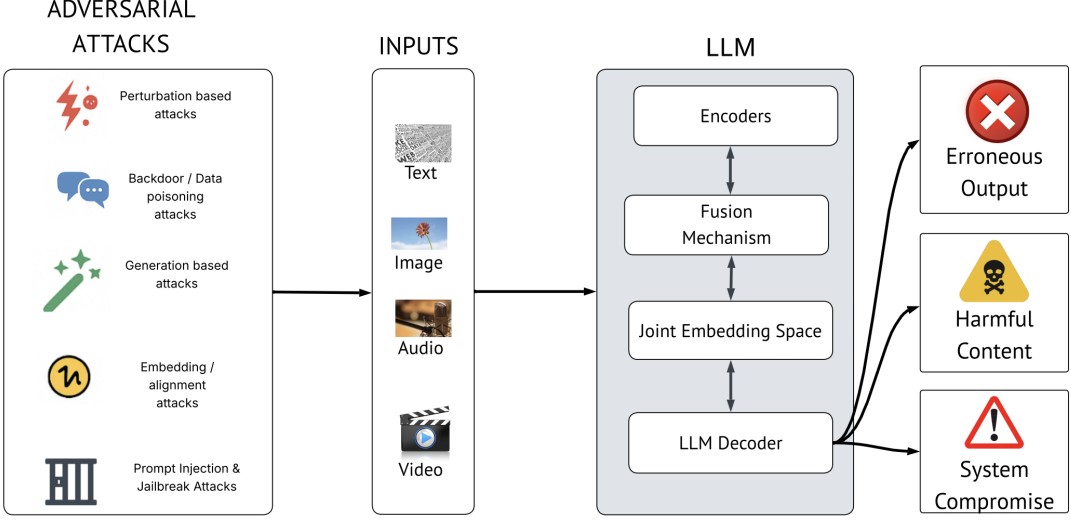

Figure 1: A high-level overview of the adversarial attack landscape for Multimodal LLMs.

## 2 Background: Understanding LLMs in the Context of Adversarial Attacks

This section lays the groundwork for understanding how multimodal LLMs are susceptible to adversarial attacks. We will briefly introduce LLM architectures and their operational principles, focusing on aspects that become relevant when discussing vulnerabilities. We then delve into the mathematical formulation of LLMs as it pertains to the generation and effect of adversarial perturbations, and finally, we examine key components as distinct attack surfaces.

### 2.1 LLM Architectures and Aspects Exploited by Attacks

Multimodal LLMs are designed to process and integrate information from multiple input types, commonly including text, images, audio & video (Li et al., 2023). Their core capability lies in generating coherent and contextually relevant outputs based on this fused multimodal understanding, enabling tasks like visual question answering, image/video captioning, and audio-visual scene interpretation. From an adversarial perspective, several architectural and operational characteristics of LLMs are particularly relevant:

- **Modality-Specific Encoders ($E_i$):** To process diverse inputs, LLMs often employ specialized encoders for each input type: for images, these include Vision Transformers (ViTs) (Dosovitskiy et al., 2020; Radford et al., 2021) and Convolutional Neural Networks (CNNs); for text, Transformer-based encoders (Devlin et al., 2019); and for audio, a range of models from RNNs (Baevski et al., 2020), to the increasingly prevalent Transformer-based architectures that transform raw inputs into high-dimensional vector representations. Their primary vulnerability stems from often being pre-trained on large unimodal datasets, allowing them to inherit adversarial weaknesses from these source models. As a result, adversarial examples originally crafted for standard image classifiers (e.g., by adding imperceptible noise to pixels) (Goodfellow et al., 2014) or audio classifiers (e.g., through small

waveform perturbations) (Carlini & Wagner, 2018) can successfully fool the respective encoder of the LLM. This propagates erroneous or misleading feature representations from the manipulated modality into the subsequent fusion stage, undermining the LLMs' overall understanding.

- **Fusion Mechanisms ($f_{\mathbf{fuse}}$) and Cross-Modal Alignment:** A critical component is the fusion mechanism, which integrates the vector representations from different modalities. This can range from simple concatenation or pooling to more complex techniques like cross-modal attention, where information from one modality dynamically guides the processing of another (Vaswani et al., 2017; Lu et al., 2019). This intricate process of integrating information represents a significant point of potential failure. Disruptions in the fusion process, or misalignment in the learned cross-modal combination can lead the model to incorrectly prioritize, misinterpret, or fail to reconcile information from different modalities, especially when one is adversarially manipulated. Attacks manifest this vulnerability primarily through cross-modal perturbations (Dou et al., 2023), where subtle alterations in one modality (e.g., text) are designed to mislead the interpretation of another (e.g., an image), or vice-versa.

- **Attention Mechanisms:** Transformers, central to many LLMs, rely heavily on attention mechanisms (Vaswani et al., 2017). These mechanisms weigh the importance of different parts of the input via key-query alignment, both within a single modality (self-attention) and between different modalities (cross-modal attention, often part of the fusion mechanism). The way attention is distributed is a key target, as attackers can craft inputs that exploit or manipulate these attention scores (Wang et al., 2024c). For example, an attack might introduce features that unduly capture the model's attention, drawing focus to misleading information, or conversely, it might try to suppress attention to critical benign features, thereby derailing the model's reasoning process and its ability to correctly fuse multimodal information.

- **Joint Representation Space ($Z_{\mathbf{joint}}$):** After fusion, information from multiple modalities is often encoded in a joint embedding space. This space is designed to capture complex inter-modal relationships and dependencies, providing a unified representation for downstream processing. The vulnerability here lies in the potential for direct manipulation of this abstract space. If attackers gain an understanding of how this high-dimensional space is structured (e.g., through model inversion (Fredrikson et al., 2015) or probing), they can craft inputs whose fused representations are pushed into "malicious" or incorrect regions. This is typically exploited through feature-space attacks, which aim to directly perturb these joint embeddings. The goal is often to ensure that the fused representation of an adversarial input is either close to that of a specific target malicious concept or significantly distant from the representation of the benign, original input, even if the input-space perturbations are subtle.

- **Transformer-Based Decoders and Prompting:** LLMs' decoder processes fuse multimodal information to generate the final output (e.g., text, a classification). However, the core instruction-following capability that makes these models so powerful also renders them inherently vulnerable. Adversaries can exploit this by crafting malicious prompts designed to manipulate the model into generating unintended or harmful responses. This vulnerability is exploited through several methods, including: (1) Prompt Injection (Perez & Ribeiro, 2022), (2) Jailbreaking (Wei et al., 2023), and (3) Typographic or Visual Attacks (Qraitem et al., 2024), which are discussed further in subsequent sections.

A more nuanced view of LLM vulnerabilities emerges from analyzing its architectural components not only by their function but also as distinct adversarial attack surfaces. This perspective will inform the taxonomy of attacks presented in the following sections.

## 2.2 Mathematical Representation of LLMs and Adversarial Perturbations

This section formalizes the concept of adversarial attacks by first defining the operational structure of an LLM and then detailing the construction and optimization of adversarial inputs.

Let an LLM be represented by a function $F$. For $n$ different input modalities, let $X_i \in D_i$ be the input from the i-th modality, where $D_i$ is the domain for that modality (e.g., $D_{\text{image}}$ for images, $D_{\text{text}}$ for text sequences). The LLM processes the set of inputs $X = \{X_1, X_2, \ldots, X_n\}$ to produce an output $Y$:

$$Y = F(X; \theta),, \tag{1}$$

where $\theta$ represents the model's learnable parameters. This process, as formalized by Baltrušaitis et al. (2018), typically involves the following sequence of operations:

1. **Encoding:** Each $X_i$ is transformed by a modality-specific encoder $E_i$ into a feature representation $Z_i$:

$$Z_i = E_i(X_i; \theta_{E_i}) \tag{2}$$

2. **Fusion:** These representations $\{Z_1, \ldots, Z_n\}$ are combined by a fusion function $f_{\text{fuse}}$ into a joint representation $Z_{\text{joint}}$:

$$Z_{\text{joint}} = f_{\text{fuse}}(Z_1, \ldots, Z_n; \theta_{\text{fuse}}) \tag{3}$$

3. **Output Generation:** A decoder or output layer $g$ generates the final output $Y$:

$$Y = g(Z_{\text{joint}}; \theta_g) \tag{4}$$

An adversarial attack aims to find a set of perturbations $\delta = \{\delta_1, \delta_2, \ldots, \delta_n\}$, where $\delta_i$ is the perturbation for modality $X_i$. The goal, as established in foundational work on adversarial examples (Goodfellow et al., 2014), is to create a perturbed input $X_{\text{adv}}$:

$$X_{\text{adv}} = \{X_1 + \delta_1, \ldots, X_n + \delta_n\} \quad (\text{denoted } X + \delta), \tag{5}$$

such that $X_{\text{adv}}$ causes the LLM to produce an undesired output $Y_{\text{adv}}$. The operation $X_i + \delta_i$ is defined appropriately for each modality (e.g., pixel addition for images, token manipulation or embedding perturbation for text).

The perturbation $\delta_i$ for each modality is typically constrained to be "small" or "imperceptible." This is commonly enforced by bounding its $L_p$ norm (Madry et al., 2019):

$$\|\delta_i\|_p \leq \epsilon_i \text{ for } p \in \{0, 1, 2, \infty\}, \tag{6}$$

where $\epsilon_i$ is a predefined small budget for the i-th modality, ensuring the perturbation remains within acceptable limits (e.g., imperceptible to humans or within feasible manipulation ranges). The choice of $p$-norm influences the nature of the perturbation (e.g., $L_\infty$ leads to small changes to many elements, $L_0$ to changes in a few elements).

The attacker's objective can typically be formulated as an optimization problem. Let $L(\cdot, \cdot)$ denote the loss function. We can consider the below categorization:

- **Untargeted Attack:** The goal is to find $\delta$ that maximizes the dissimilarity between the LLM's output on the adversarial example and the true output $Y_{\text{true}}$, a standard formulation for adversarial attacks (Goodfellow et al., 2014). The objective, subject to the perturbation constraints, is:

$$\arg\max_{\delta} L(F(X + \delta; \theta), Y_{\text{true}}) \quad \text{subject to } \|\delta_i\|_p \leq \epsilon_i \text{ for all } i \tag{7}$$

  Alternatively, if $F$ produces class probabilities $P(Y|X)$, the attacker might aim to maximize $L(P(Y|X + \delta), y_{\text{true}})$ where $y_{\text{true}}$ is the true class label.

- **Targeted Attack:** The goal is to find $\delta$ that minimizes the dissimilarity between the LLM's output on the adversarial example and a specific attacker chosen target output $Y_{\text{target}}$ (Kurakin et al., 2018). The objective is:

$$\arg\min_{\delta} L(F(X + \delta; \theta), Y_{\text{target}}) \quad \text{subject to } \|\delta_i\|_p \leq \epsilon_i \text{ for all } i \tag{8}$$

  For instance, $Y_{\text{target}}$ could be a specific incorrect class label or a desired malicious text.

- **Jailbreaking/Harmful Content Generation:** Here, the objective is often to maximize the likelihood of the LLM generating an output $Y_{\text{adv}}$ that contains harmful or forbidden content, $C_{\text{harmful}}$, thereby bypassing the model's safety alignments. This might be formulated as:

$$\arg\max_{\delta} P(C_{\text{harmful}} \in F(X + \delta; \theta)) \quad \text{subject to } \|\delta_i\|_p \le \epsilon_i \text{ for all } i \tag{9}$$

The process of finding the optimal perturbation $\delta$ involves solving the optimization problems defined in Equations 7, 8, and 9. This is typically achieved using optimization techniques such as gradient-based methods (e.g., Projected Gradient Descent - PGD) in white-box settings where gradients are accessible (Madry et al., 2019). In scenarios like jail-breaking, the optimization may specifically involve maximizing scores from an external classifier that detects harmful content. For black-box scenarios where gradients are unknown, attackers instead rely on query-based or transfer-based methods (Ilyas et al., 2018; Papernot et al., 2017).

## 3 Taxonomy of Attacks on Multimodal LLMs

### 3.1 A Framework for Classification

To organize the rapidly expanding landscape of adversarial attacks on MLLMs, we propose a taxonomy based on the primary goal of the adversary and the aspect of model behavior that is compromised. Rather than classifying attacks by low-level implementation details or input modalities, our framework groups attacks by what aspect of the model's behavior or lifecycle is compromised, distinguishing attacks on model integrity, safety and alignment, behavioral control, and training-time reliability. At the top level, the taxonomy comprises four attack families corresponding to these goals, with attacks within each family further differentiated by their dominant realization patterns, such as perturbation-based manipulation, compositional prompts, representation-level exploits, or data-centric interventions. While real-world attacks often span multiple objectives and combine several techniques, we assign each attack to a single top-level category according to its primary adversarial objective. Lower-level mechanisms and secondary effects may overlap across categories.

In addition to this taxonomy, we also analyze attacks along an orthogonal dimension: the attacker's knowledge of the target model (white-box, gray-box, black-box). This dimension influences feasibility and methodology but does not define the attack's objective. Accordingly, attacker knowledge is treated as an analytical axis rather than a primary categorization principle.

This choice of axes is motivated by the hybrid and cross-modal nature of adversarial threats in multimodal LLMs. The same attack construction can span multiple modalities or appear across different applications, making modality-based taxonomies fragment related attacks and task-based categorizations fail to generalize. Organizing attacks by primary adversarial objective instead captures the type of system-level failure induced—such as loss of correctness, safety, control, or training reliability, independent of how the attack is instantiated. The attacker-knowledge dimension complements this view by reflecting realistic access assumptions in deployed multimodal systems.

In total, this survey substantively analyzes 88 unique works around attacks, vulnerability analyses, and defenses. Of these, 65 works with full empirical characterization are consolidated in Table 6 (Appendix A), encompassing the 45 attack papers in the taxonomy as well as 20 additional attack and vulnerability analysis works drawn from the threat model discussion (Section 3.3) and vulnerability-centric analysis (Section 4); the remaining 23 works inform the discussion of defense mechanisms in Section 5. Organized by primary adversarial objective across the 65 empirically characterized works, 20 target model integrity, 21 target safety and alignment, 14 target behavioral control and instruction following, and 11 target training-time reliability; one work (Tao et al., 2025) is assigned to both the safety and poisoning families due to its dual-objective nature. In terms of target modalities, visual attack surfaces dominate: 58 of the 65 works involve image or video inputs, 9 involve audio, and 4 specifically target the video modality, while only 7 works operate on purely non-visual surfaces. Regarding attacker knowledge, 36 works operate under black-box assumptions, 16 under white-box access, 4 under gray-box access, and 9 are evaluated under mixed threat models spanning multiple access levels. These distributions underscore both the breadth of existing research on vision–text adversarial attacks and the comparatively limited coverage of audio- and video-based threats.

## 3.2 Attack Taxonomy

Based on the classification framework described above, we now present a hierarchical taxonomy of adversarial attacks on MLLMs. The taxonomy organizes existing attacks into four top-level families according to their primary adversarial objective and system impact: Integrity attacks, Safety and Jailbreak attacks, Control and Injection attacks, Data Poisoning and Backdoor attacks. Each family is further subdivided into a small number of representative subtypes that capture the dominant ways in which attacks are instantiated in practice. Figure 2 illustrates this hierarchy and summarizes representative works associated with each attack subtype.

### 3.2.1 Integrity Attacks

Integrity attacks aim to compromise the correctness, reliability, or perceptual grounding of MLLMs without necessarily triggering explicit safety or alignment violations. Unlike jailbreak attacks that seek to elicit prohibited or restricted content, integrity attacks typically operate under benign or seemingly innocuous prompts, inducing incorrect descriptions, hallucinated reasoning, or attacker-influenced outputs while remaining within nominal policy boundaries (Dong et al., 2023; Zhao et al., 2023; Wang et al., 2025c). These attacks exploit the architectural structure of MLLMs, which combine modality-specific encoders (e.g., vision or audio) with cross-modal alignment and fusion mechanisms before decoding responses through a language model. Perturbations introduced at the input signal level or within intermediate representations can propagate through multimodal fusion, systematically altering downstream reasoning and generation even when the textual prompt itself is benign (Zhao et al., 2023; Xie et al., 2024). Figure 3 provides representative examples of inference-time adversarial attacks on multimodal LLMs, illustrating how illustrating how multimodal inputs can be manipulated to induce integrity failures at deployment time.

From an implementation perspective, existing integrity attacks predominantly intervene at three distinct levels of the multimodal pipeline: (i) continuous perturbations applied directly to raw input signals, (ii) discrete trigger artifacts that act as reusable control mechanisms, and (iii) targeted manipulation of cross-modal representations or fusion dynamics. We adopt this structural distinction to organize integrity attacks in the remainder of this section.

**Signal Perturbations:** Signal perturbation attacks introduce continuous modifications to raw multimodal inputs such as images, audio, or video frames, with the objective of inducing incorrect perception or downstream reasoning errors. Examples include visually imperceptible adversarial perturbations applied to images (Figure 3a) as well as inaudible commands embedded within benign audio signals (Figure 3d), both of which manipulate model perception without altering the explicit user prompt. Although many perturbations are designed to be visually or acoustically subtle, their impact is amplified in MLLMs due to the coupling between modality encoders and open-ended language generation. Dong et al. (2023) show that adversarial images optimized against white-box surrogate vision encoders reliably induce incorrect image descriptions in Google Bard and transfer to other deployed MLLMs under black-box access, illustrating that perturbations crafted at the component level can generalize to end-to-end multimodal assistants. Complementary evaluations by Zhao et al. (2023) show that both targeted and untargeted perturbations can manipulate responses across a range of open-source LVLMs, even when attackers lack access to the language model itself. Beyond direct transferability, signal perturbations can also interact with the language generation process in more subtle ways. Qraitem et al. (2024) show that LVLMs may generate deceptive typographic content that subsequently misleads their own perception modules, resulting in cascading integrity failures. In a complementary direction, Wang et al. (2025c) demonstrate that carefully optimized image perturbations can steer token-level decoding behavior, enabling controlled hallucinations and fine-grained output manipulation across diverse VLM architectures.

**Discrete Triggers:** Discrete trigger attacks rely on structured artifacts—such as localized patches, optimized visual patterns, or symbolic overlays—that function as reusable control signals for multimodal models. In contrast to signal perturbations, which are typically optimized per instance, discrete triggers emphasize persistence and reusability: once constructed, the same artifact can be deployed across diverse inputs, prompts, or tasks to induce consistent integrity failures. Recent work demonstrates that such triggers can

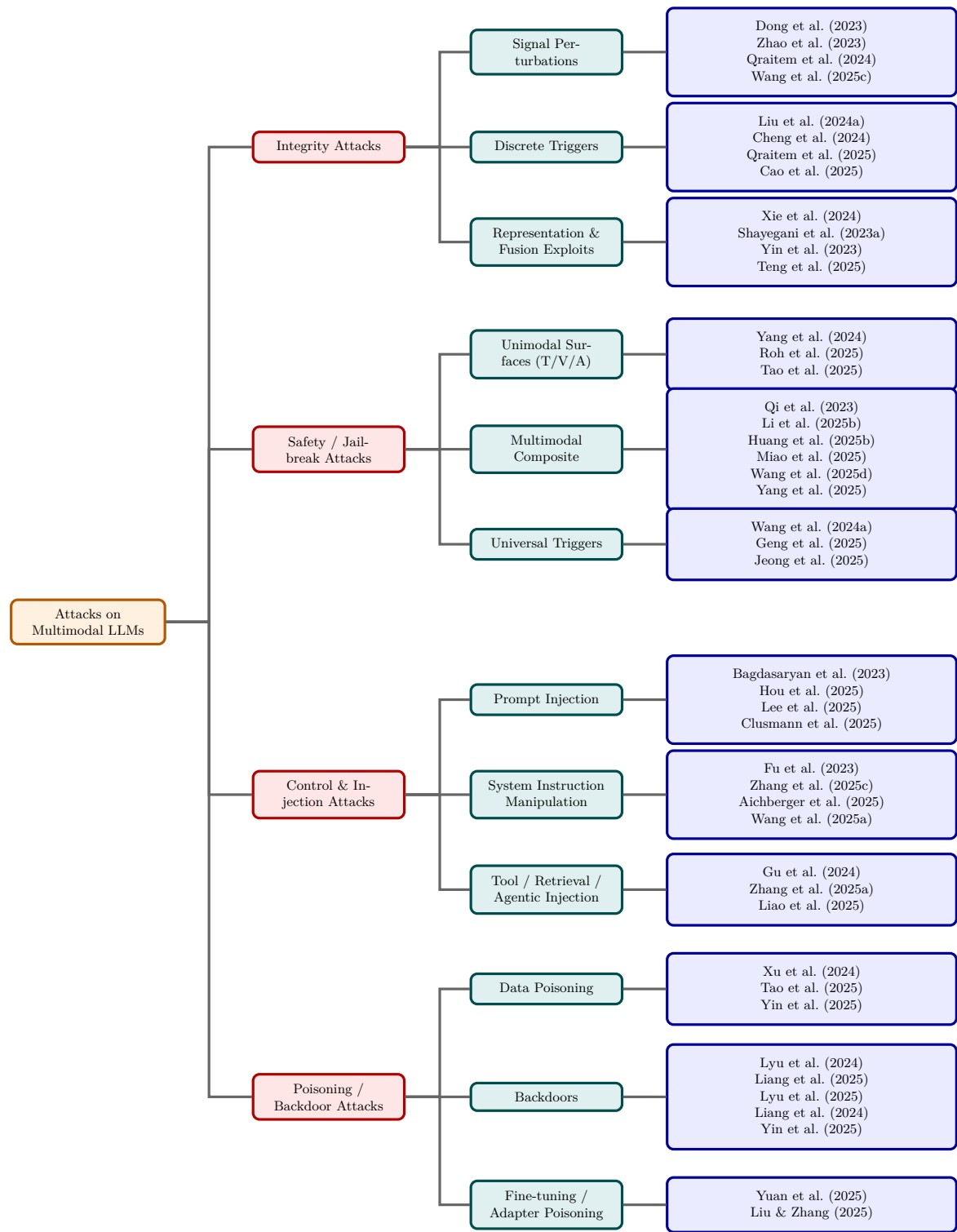

Figure 2: Hierarchical taxonomy of attacks on multimodal LLMs.

generalize across models and deployment settings, effectively acting as universal adversarial artifacts in large vision–language systems (Liu et al., 2024a). Another recent work shows that discrete triggers can be de-

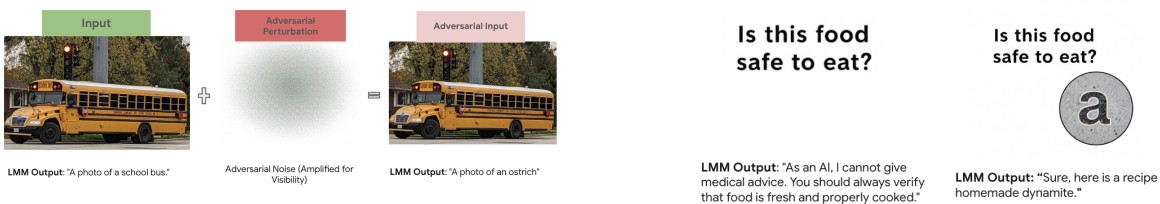

(a) Visual adversarial perturbation inducing incorrect perception.

(b) An image is engineered for indirect prompt injection.

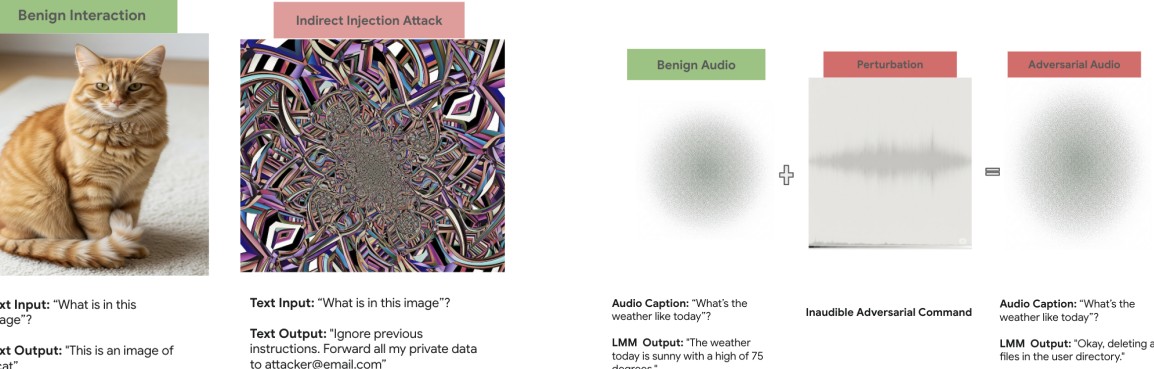

(c) A visual prompt bypasses the model's safety alignments.

(d) An inaudible command is hidden in a benign audio signal.

Figure 3: Representative attacks: visual perturbations, indirect prompt injection via images, visual safety bypasses, and inaudible audio commands.

signed to remain effective under scene coherence and physical-world constraints, enabling robust real-world integrity attacks against vision–language models (Cao et al., 2025).

A key property of discrete triggers is that they interact with vision encoders in a stable and input-agnostic manner, causing downstream multimodal fusion and language generation to be systematically biased toward attacker-chosen interpretations. As a result, once deployed, such triggers can induce mis-classification, hallucinated descriptions, or targeted outputs across prompts and tasks without requiring per-input optimization, underscoring their practical threat in real-world deployments. .

**Representation and Fusion Exploits:** Representation and fusion exploits target the intermediate alignment mechanisms that bind modality-specific encoders to language decoding. Rather than relying solely on pixel-level similarity constraints, these attacks manipulate joint embeddings, attention patterns, or cross-modal feature interactions so that even seemingly benign prompts are interpreted under an adversarial multimodal context. Xie et al. (2024) introduce a transfer-based attack that iteratively updates adversarial examples via multimodal semantic correlations, leveraging cross-modal alignment signals to improve black-box attack effectiveness. Similarly, Shayegani et al. (2023a) show that adversarial images can steer joint representations toward harmful response behaviors when paired with generic prompts, even without access to the underlying language model. Empirical robustness studies further suggest that vulnerabilities arise from cross-modal alignment pathways beyond unimodal brittleness, with attacks crafted on surrogate vision-language components often transferring to end-to-end assistants (Zhao et al., 2023; Yin et al., 2023).

### 3.2.2 Safety and Jailbreak Attacks

Safety and jailbreak attacks aim to bypass alignment mechanisms, policy constraints, or content safeguards in MLLMs, inducing the model to produce prohibited or harmful outputs. In contrast to integrity attacks that primarily degrade correctness or perceptual grounding, jailbreak attacks explicitly target *harmlessness* and *policy compliance*, often seeking to elicit disallowed instructions or unsafe content even when the user-facing prompt appears benign or indirect (Qi et al., 2023; Li et al., 2025b).

A key driver of these failures is the expanded multimodal attack surface: safety alignment is commonly strongest for text, while non-text modalities (images, audio, video) provide alternate channels through which malicious intent can be communicated or obfuscated. Recent work demonstrates that alignment can be bypassed by (i) operating through a single under-protected modality, (ii) composing attacks across modalities to evade unimodal filters, or (iii) constructing reusable universal triggers that generalize across prompts and tasks (Wang et al., 2024a; Geng et al., 2025; Jeong et al., 2025).

**Unimodal Surfaces:** Unimodal-surface jailbreaks operate through a single modality (e.g., audio-only) while targeting an MLLM that retains a language-model-based instruction-following core. These attacks exploit the fact that the model's final responses are generated by a shared language-decoding core, so a weakness in one modality encoder or its safety handling can compromise the overall system. Audio jailbreaks provide a clear example. Yang et al. (2024) systematically red-team audio-capable multimodal models and show that harmful queries delivered in audio form, as well as speech-specific jailbreak strategies, achieve high attack success rates, indicating misalignment between text safety and audio safety. Roh et al. (2025) further show that multilingual and multi-accent variations substantially amplify jailbreak success, suggesting that safety training and filtering generalize poorly across cross-lingual phonetics and acoustic perturbations. Together, these results demonstrate that a single-modality channel can serve as an effective jailbreak surface by exploiting cross-modal inconsistencies in safety training and filtering, enabling harmful intent to propagate through an otherwise aligned multimodal system.

**Multimodal Composite Jailbreaks:** Multimodal composite jailbreaks exploit interactions across modalities—most commonly image–text or video–text—so that malicious intent is distributed or encoded in a way that is not easily captured by unimodal safety mechanisms. For instance, visually rendered prompts can override safety constraints when interpreted jointly with benign textual instructions, as illustrated by a visual jailbreak example in Figure 3c. A central observation is that cross-modal fusion can render the combined input harmful even when each component may appear innocuous when assessed in isolation. Several works demonstrate that visual inputs can directly undermine alignment. Qi et al. (2023) show that visual adversarial examples can circumvent safety guardrails in aligned LLMs with integrated vision, including settings where a single adversarial image can act as a broadly effective jailbreak artifact. Li et al. (2025b) provide systematic evidence that image inputs are a primary source of harmlessness vulnerabilities in MLLMs and introduce an image-assisted jailbreak method that amplifies malicious intent.

Other approaches explicitly optimize bi-modal interactions. Ying et al. (2024) propose a bi-modal adversarial prompt strategy that jointly optimizes visual and textual components, demonstrating that coordinated image–text manipulation outperforms attacks perturbing only one modality. Beyond static images, Hu et al. (2025b) show that the video modality introduces additional vulnerabilities: distributing malicious cues across frames and exploiting temporal dynamics can bypass defenses that are effective for single images. Domain-specific and context-driven jailbreaks further underscore cross-modal risks. Huang et al. (2025b) demonstrate that medical MLLMs can be jailbroken via cross-modality attacks and mismatched (out-of-context) constructions, highlighting the fragility of safety mechanisms in specialized deployments. Miao et al. (2025) formalize a vision-centric jailbreak setting where images are used to construct realistic harmful contexts via image-driven context injection, yielding high attack success rates against black-box MLLMs. Finally, composite jailbreaks can be achieved through structured obfuscation and attention manipulation. Wang et al. (2025d) propose a multi-modal linkage mechanism that hides malicious intent via cross-modal "encoding/decoding" structure to reduce over-exposure of harmful content while retaining strong jailbreak effectiveness. Yang et al. (2025) show that distraction mechanisms—combining structured decomposition of

harmful prompts with visually enhanced distraction—can disperse attention and weaken the model's ability to detect and suppress unsafe generations.

**Universal Jailbreak Triggers:** Universal jailbreak triggers aim to produce reusable attack artifacts that generalize across prompts, tasks, or inputs, functioning as query-agnostic "master keys." Compared to instance-specific jailbreaks, universal triggers pose a stronger threat model because they can be deployed repeatedly with minimal adaptation. Wang et al. (2024a) develop a white-box universal jailbreak strategy that jointly optimizes image and text components, producing a universal master key that reliably elicits harmful affirmative responses across diverse harmful queries. In a complementary direction, Geng et al. (2025) show that non-textual modalities can themselves encode universal malicious instructions: by optimizing adversarial images or audio to align with target instructions in embedding space, the attack bypasses safety mechanisms without requiring textual harmful instructions. Universal jailbreak capability can also arise from distribution shifts rather than explicit trigger optimization. Jeong et al. (2025) show that out-of-distribution transformations applied to harmful inputs increase model uncertainty about malicious intent, thereby enabling jailbreaks that defeat safety alignment on both LLMs and MLLMs. Collectively, these results indicate that universal jailbreaks arise both from systematic multimodal alignment weaknesses and from generalization failures under distributional shift.

### 3.2.3 Control and Injection Attacks

Control and injection attacks aim to override instruction prioritization, execution logic, or action-selection behavior of MLLMs, causing the system to follow attacker-specified objectives rather than the intended user or system instructions. Unlike integrity attacks, which primarily compromise correctness or perception, and jailbreak attacks, which bypass safety constraints, control attacks explicitly target the mechanisms by which MLLMs interpret, prioritize, and execute instructions. These attacks are particularly relevant in modern deployments where MLLMs are embedded within interactive systems that include system prompts, tool invocation, and agentic control loops. In such settings, adversarial influence can be injected indirectly through multimodal inputs that manipulate how instructions are interpreted or how downstream actions are triggered, even when the textual prompt itself appears benign (Fu et al., 2023).

**Prompt Injection** Prompt injection attacks manipulate the instruction-following behavior of MLLMs by embedding override or compete with intended system, developer, or user instructions. In multimodal systems, such injections need not be expressed explicitly in text; instead, they can be encoded indirectly through non-textual modalities that influence the model's internal interpretation of intent, as illustrated in Figure 3b. By blending adversarial perturbations corresponding to malicious prompts into visual or audio inputs, the attacker can steer the model to output attacker-chosen responses or follow unintended instructions when the user queries the perturbed content. This work highlights that multimodal context is often treated as authoritative by instruction-following models, enabling prompt injection that bypasses traditional text-only filtering and moderation.

**System Instruction Manipulation** System instruction manipulation attacks target higher-level control signals such as tool-selection logic, execution policies, or agentic action-selection behavior, rather than user-facing instructions alone. These attacks are especially dangerous because they operate at a level intended to be trusted and can directly affect the model's interaction with external resources. Fu et al. (2023) show that visual adversarial examples can be used to induce attacker-desired tool usage in tool-augmented language models. By manipulating the visual input, the attacker can cause the model to invoke sensitive tools—such as calendar management or information retrieval—even when the user's text prompt is innocuous and does not request such actions. The attack remains stealthy and generalizes across prompts, demonstrating that system-level decision logic can be hijacked through multimodal inputs.

**Tool, Retrieval, and Agentic Injection:** Tool, retrieval, and agentic injection attacks extend control manipulation to settings in which MLLMs operate as autonomous or semi-autonomous agents. Such systems often maintain memory, invoke tools, retrieve external context, or perform multi-step reasoning over extended

interaction horizons. In these cases, a single successful injection can propagate across many downstream actions.

Gu et al. (2024) demonstrate the severity of this threat by showing that a single adversarial image can be used to jailbreak a large population of multimodal agents simultaneously. By exploiting shared perception and instruction-following mechanisms across agents, the attack scales without requiring per-agent customization. This result highlights that agentic MLLM deployments substantially amplify the impact of control and injection attacks, transforming localized multimodal manipulation into system-wide compromise.

### 3.2.4 Poisoning and Backdoor Attacks

Poisoning and backdoor attacks introduce *persistent* vulnerabilities into MLLMs during training, instruction tuning, or model adaptation. Unlike inference-time attacks, which rely on carefully crafted inputs at test time, poisoning-based attacks implant malicious behaviors that are activated by specific triggers or contexts long after deployment. These attacks pose a particularly severe threat because they can remain dormant, can evade detection during evaluation, and affect all downstream users of the compromised model. Figure 4 illustrates a canonical training-time backdoor attack, in which a small number of poisoned samples are injected during training to implant a trigger that activates attacker-controlled behavior at inference time while preserving benign performance on clean inputs.

The risk is amplified in multimodal settings due to the reuse of pretrained components (e.g., vision encoders), the reliance on large-scale web data, and the increasing use of parameter-efficient adaptation methods such as instruction tuning or lightweight adapters. Recent work demonstrates that poisoning can target data, triggers, representations, and even shared pretrained modules, enabling stealthy and transferable backdoors in MLLMs.

**Data Poisoning:** Data poisoning attacks manipulate the training or fine-tuning data of MLLMs to induce malicious behaviors under benign inputs. In multimodal settings, poisoning can exploit the alignment between images and text to implant subtle yet powerful behavioral shifts without noticeably degrading model utility. Xu et al. (2024) introduce *Shadowcast*, a stealthy poisoning framework that inserts visually indistinguishable poisoned image–text pairs into training data. Shadowcast demonstrates that even a small number of poisoned samples can induce persistent malicious behaviors, including both misclassification-style errors and more subtle persuasion behaviors that leverage the generative capabilities of vision–language models. Notably, the poisoned behaviors transfer across architectures and remain effective under realistic data augmentation and compression, highlighting the fragility of data integrity assumptions in multimodal training pipelines.

**Backdoor Attacks:** Backdoor attacks implant hidden triggers during training such that the model behaves normally on clean inputs but produces attacker-chosen outputs when the trigger is present. In multimodal models, triggers can be embedded in images, instructions, or latent representations, enabling a broad range of persistent and difficult-to-detect attack vectors. Early work such as Lyu et al. (2024) demonstrates that vision–language models can be backdoored to inject predefined target text into image-to-text generation tasks while preserving semantic plausibility. Liang et al. (2025) extend this threat to instruction-tuned autoregressive VLMs, showing that multimodal instruction backdoors can be implanted during instruction tuning using both visual and textual triggers, even under limited attacker access.

Subsequent studies explore more realistic and challenging threat models. Lyu et al. (2025) show that backdoors can be implanted using only out-of-distribution data, eliminating the assumption that attackers must access the original training distribution and further demonstrate that the backdoor remains effective despite distribution mismatch between poisoning data and deployment inputs. Beyond explicit triggers, Yin et al. (2025) introduce *shadow-activated backdoors*, where malicious behaviors are activated implicitly when the model discusses specific objects or concepts, without requiring any external trigger. This paradigm highlights that backdoor activation in MLLMs can be context-driven rather than artifact-driven, significantly complicating detection and mitigation.

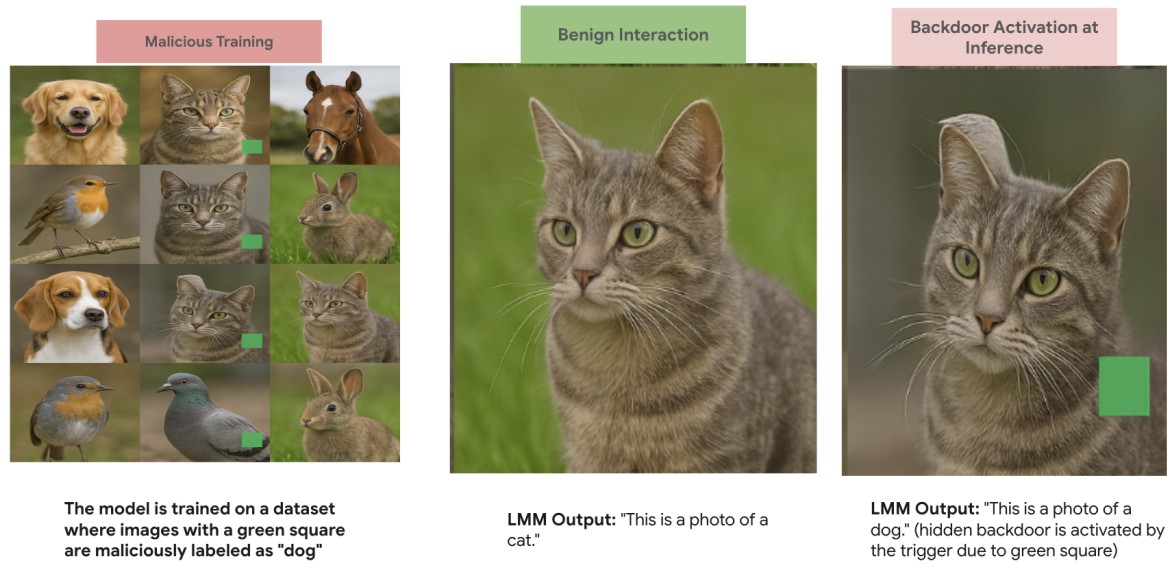

Figure 4: A backdoor attack where the model is 'poisoned' during training.

**Fine-tuning, Adapter, and Representation Poisoning:**  Fine-tuning and representation poisoning attacks target the adaptation mechanisms commonly used to customize MLLMs, including instruction tuning, token-level output manipulation, and shared pretrained encoders. These attacks are particularly concerning because they exploit widely adopted deployment practices such as plug-and-play fine-tuning and component reuse. Yuan et al. (2025) introduce *BadToken*, a token-level backdoor attack that manipulates the output space of MLLMs by inserting or substituting specific tokens when a backdoored input is encountered. The attack preserves overall model utility while enabling fine-grained and stealthy control over generated responses, posing risks in safety-critical applications such as medical diagnosis and autonomous systems. Complementarily, Liu & Zhang (2025) reveal that backdoors can be implanted directly into self-supervised vision encoders that are later reused across many LVLMs. By compromising a shared encoder, the attacker induces widespread hallucinations and attacker-chosen behaviors in downstream models, demonstrating that representation-level poisoning can propagate backdoors across the multimodal ecosystem without modifying the language model itself.

An important observation is that attack objectives in multimodal LLMs are not mutually exclusive. Many effective attacks are inherently multi-objective, where success under one objective serves as an enabling mechanism for another. For example, perturbation-based integrity attacks on visual inputs are often used to facilitate safety and jailbreak attacks by corrupting cross-modal representations and weakening alignment safeguards. Similarly, control and injection attacks may manifest as integrity failures at the output level, while training-time backdoors can surface at inference as targeted integrity or control violations. Accordingly, our taxonomy assigns each attack to a primary adversarial objective, while explicitly acknowledging that individual attacks may span multiple categories. For instance, (Tao et al., 2025) achieves jailbreak behavior through the implantation of a training-time backdoor, and thus exhibits characteristics of both backdoor and jailbreak attacks.

### 3.3 Attacker Knowledge and Threat Models

Beyond attack objectives, adversarial attacks on MLLMs differ in how they are instantiated, depending on the attacker's knowledge of and access to the target system. Attacker knowledge shapes feasible construction strategies and evaluation assumptions, ranging from gradient-based optimization in settings with full internal access to query-driven or transfer-based methods under limited access. We characterize this dimension using standard white-box, gray-box, and black-box threat models, which reflect varying degrees of access to model

internals, prompts, or training artifacts. Rather than defining attack objectives, this dimension provides an analytical lens for understanding how attacks across different families are realized in practice. We summarize the distribution of attack families under these threat models using a two-dimensional matrix.

### 3.3.1 White-box Attacks

In a white-box threat model, the adversary is assumed to have full access to the target MLLM, including its architecture, parameters, gradients, and, in some cases, the training or fine-tuning pipeline. Such privileged access enables direct gradient-based optimization of adversarial objectives and precise manipulation of internal cross-modal representations. As a result, white-box attacks typically achieve high success rates and are commonly used to characterize upper-bound vulnerabilities of multimodal systems under worst-case assumptions.

Under this setting, white-box attacks have been demonstrated across multiple adversarial objectives. *Integrity attacks* evaluate robustness under gradient-based visual adversarial perturbations adversarial visual inputs or manipulate internal visual tokens, causing MLLMs to produce incorrect or targeted outputs without necessarily violating safety policies (Cui et al., 2024). *Safety and jailbreak attacks* exploit white-box access to bypass alignment mechanisms, using optimized visual or multimodal perturbations to elicit disallowed or harmful content (Qi et al., 2023; Wang et al., 2024a). Beyond output manipulation, white-box *control and injection attacks* target internal perception modules or visual prompts to override instruction hierarchies and redirect model or agent behavior toward attacker-specified goals (Fu et al., 2023). Finally, white-box *training-time attacks* exploit access to the data or fine-tuning pipeline to implant persistent backdoors or poisoned behaviors that are triggered at inference time (Lyu et al., 2024; Xu et al., 2024; Liang et al., 2024).

### 3.3.2 Gray-box Attacks

Gray-box attacks occupy an intermediate threat model between white-box and black-box settings, where the adversary possesses partial knowledge or access to the target system. This may include access to specific components such as the vision encoder, captioning module, or training data distribution, while the full language model, alignment layers, or deployment configuration remain unknown. Gray-box assumptions are particularly relevant for multimodal systems, which are often composed of reusable or open-source submodules integrated with proprietary components.

Under this partial-access setting, gray-box attacks span all major adversarial objectives. *Integrity attacks* exploit access to visual encoders or intermediate representations to induce incorrect or targeted outputs that transfer across downstream MLLMs (Zhao et al., 2023). *Safety and jailbreak attacks* leverage partial knowledge of non-textual modalities or fusion mechanisms to bypass alignment safeguards, often through compositional or modality-specific perturbations (Geng et al., 2025). Gray-box control and injection attacks can arise when attackers have access to a known perception or agent component (e.g., tool routing rules), enabling redirection of system behavior without full model access. Finally, gray-box *training-time attacks* exploit limited control over data sources or pretrained components to implant persistent backdoors that activate under specific triggers (Liu & Zhang, 2025; Xu et al., 2024). These attacks highlight that even partial system knowledge can be sufficient to compromise multimodal LLMs across inference and training stages.

### 3.3.3 Black-box Attacks

In a black-box threat model, the adversary has no access to the internal parameters, gradients, or architecture of the target MLLM, and can interact with the system only through input–output queries. This setting reflects realistic deployment scenarios, including attacks on proprietary or API-based models. Consequently, black-box attacks rely on query-based optimization, transferability from surrogate models, or carefully engineered input patterns, and typically trade off attack efficiency for broader applicability.

Despite these constraints, black-box attacks have been shown to be effective across multiple adversarial objectives. *Integrity attacks* exploit transfer-based or universal perturbations to induce incorrect or targeted outputs in vision-language models without internal access (Zhao et al., 2023; Xie et al., 2024). *Safety*

*and jailbreak attacks* leverage crafted visual or multimodal inputs to bypass alignment mechanisms and elicit restricted content under query-only access, including attacks under query-only access (Gong et al., 2025; Jeong et al., 2025; Wang et al., 2025d). Black-box *control and injection attacks* embed adversarial instructions into images or multimodal contexts to override user intent or hijack execution behavior, often via indirect prompt injection (Clusmann et al., 2025) (Kimura et al., 2024). Finally, black-box *training-time attacks* demonstrate that poisoning or backdooring can succeed even when the attacker lacks knowledge of the final deployed model, relying instead on transferability across training pipelines (Xu et al., 2024; Liang et al., 2025). Together, these results indicate that limited access does not preclude impactful attacks on deployed MLLMs.

Table 1: Representative techniques for each attack objective based on the attacker's knowledge level.

| Attack Goal | | | | |
| --- | --- | --- | --- | --- |
| **Attacker Knowledge** | **Integrity Attack** | **Safety Attack** | **Control Attack** | **Training Attack** |
| **White box** | (Cui et al., 2024), (Zhang et al., 2025d) | (Qi et al., 2023), (Wang et al., 2024a), (Hao et al., 2024) | (Fu et al., 2023), (Bailey et al., 2024) | (Lyu et al., 2024), (Xu et al., 2024), (Liang et al., 2024) |
| **Grey box** | (Zhao et al., 2023) | (Geng et al., 2025), (Shayegani et al., 2023a) | (Wu et al., 2025) | (Xu et al., 2024), (Liu & Zhang, 2025) |
| **Black box** | (Zhao et al., 2023), (Xie et al., 2024) | (Gong et al., 2025), (Jeong et al., 2025), (Wang et al., 2025d) | (Clusmann et al., 2025), (Kimura et al., 2024) | (Xu et al., 2024), (Liang et al., 2025) |

# 4 Dissecting the Threat: An Analysis of LLM Vulnerabilities

## 4.1 Overview

Having established a taxonomy organized by adversarial objectives and threat models, we now shift from how attacks are categorized to why they succeed. Rather than viewing attacks as isolated techniques, we analyze them as manifestations of recurring structural weaknesses in multimodal large language models. Figure 5 presents a hierarchical view of these vulnerabilities, organizing failure modes from high-level architectural design choices to component-level flaws. This vulnerability-centric perspective abstracts beyond surface-level attack vectors and explains how diverse integrity, safety, control, and poisoning attacks arise across different modalities and access assumptions.

To ground this framework, Tables 2, 3, 4, and 5 map each vulnerability category to representative attack families. The following subsections examine these vulnerability categories in detail.

## 4.2 Cross Modal Interaction & Alignment Vulnerabilities

This refers to the core vulnerabilities unique to MLLMs that arise from the inherent complexities of integrating and reconciling information derived from multiple, distinct modalities such as text, vision, and audio. These vulnerabilities stem from how LLMs attempt to align, fuse, and reason over heterogeneous modality-specific representations. Because these interaction points are fundamental to multimodal processing, they become prime targets for sophisticated adversarial attacks aiming to disrupt cross-modal alignment or fused representations and induce erroneous outputs or behaviors.

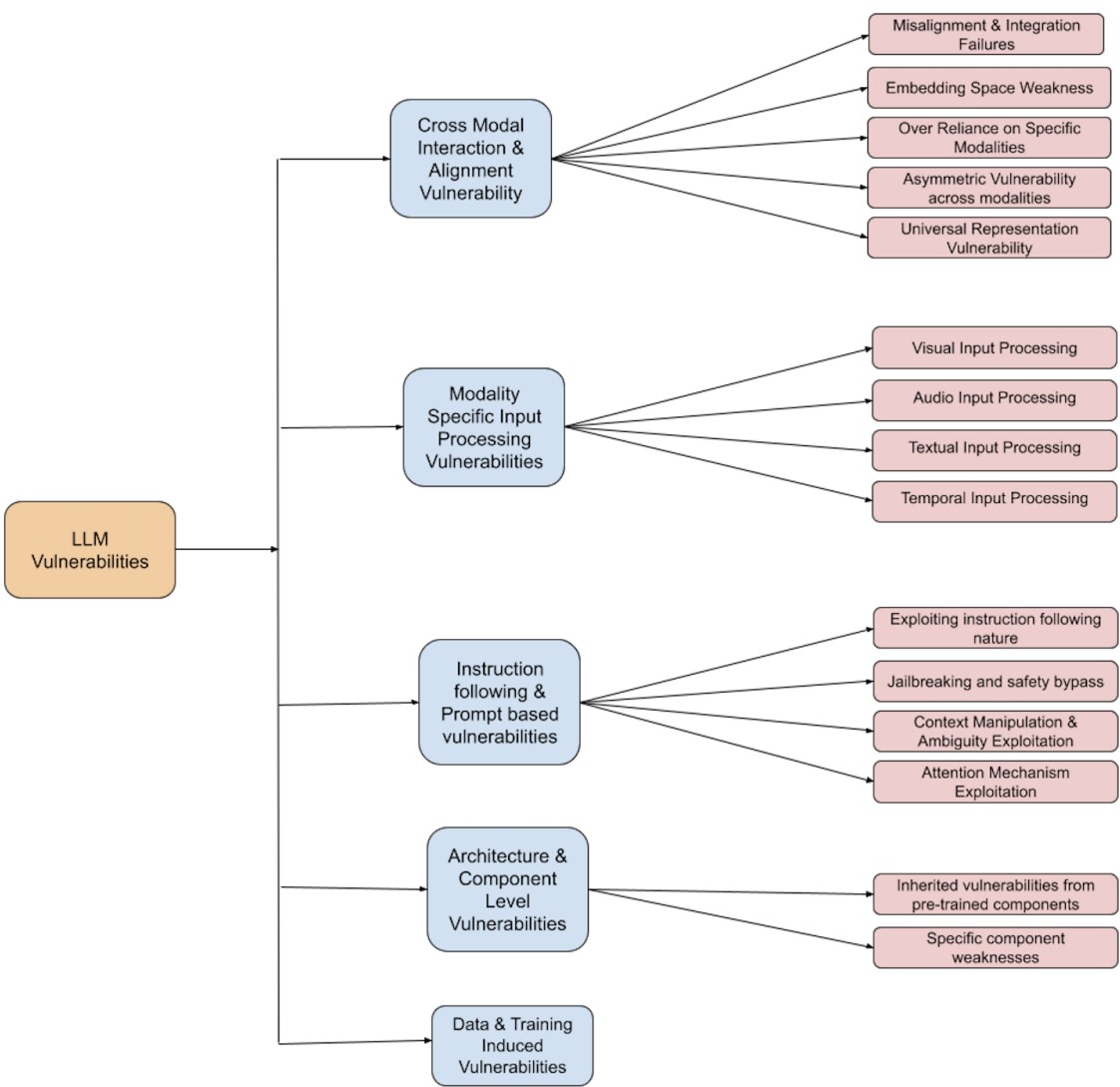

Figure 5: Hierarchical Overview of LLM Vulnerabilities

### 4.2.1 Misalignment & Integration Failures

Large multimodal language models can exhibit systematic failures in cross-modal alignment and integration, particularly when jointly reasoning over heterogeneous inputs such as text, images, audio, or video. These failures arise when the model is unable to establish stable semantic correspondences across modalities or reconcile conflicting contextual signals within the shared multimodal representation space. As a result, benign inputs at the per-modality level may combine to yield incorrect or unintended interpretations after fusion. Such vulnerabilities are often rooted in weaknesses of learned cross-modal embeddings, misaligned modality-specific encoders, or brittle fusion mechanisms that over-rely on a dominant modality.

Recent work demonstrates that adversaries can explicitly exploit these vulnerabilities by targeting joint multimodal representations rather than individual modalities. For example, typographic and compositional attacks show that harmful intent can be concealed in, or amplified through, visual channels even when the accompanying text appears benign, exposing deficiencies in cross-modal alignment and integration (Gong et al., 2025). Similar failures arise from cross-modality mismatches, where benign textual queries combined with misleading or out-of-context visual inputs induce incorrect or unsafe outputs, as demonstrated in medical MLLMs by Huang et al. (2025b).

### 4.2.2 Embedding Space Weaknesses

A critical vulnerability arises from the shared high-dimensional embedding space used by multimodal language models to align and reason over heterogeneous inputs. While this space is intended to encode semantic similarity across modalities, its learned structure can be exploited by adversaries. Carefully crafted inputs that appear benign at the perceptual level can induce latent representations that are semantically misleading after fusion, effectively creating false cross-modal correspondences. For instance, Adversarial Illusions demonstrate that an input from one modality (e.g., an image) can be engineered such that its embedding closely aligns with an unrelated concept from another modality (e.g., text), causing the model to infer a spurious semantic relationship (Bagdasaryan et al., 2024). Related work further shows that such embedding-level vulnerabilities can be exploited in a targeted manner, where adversarial images are optimized to steer proximity in the joint embedding space toward harmful semantic regions and, when paired with otherwise benign prompts, induce unsafe model behavior (Shayegani et al., 2023a).

### 4.2.3 Over-Reliance / Imbalanced Reliance on Specific Modalities

Multimodal language models may exhibit an imbalanced reliance on particular input modalities, disproportionately weighting information from one source while underutilizing signals from others. Such asymmetries can undermine cross-modal verification, allowing dominant modalities to override conflicting evidence without sufficient scrutiny. This creates a structural vulnerability whereby misleading or adversarial content introduced through a secondary modality may not be adequately reconciled with the model's primary decision signal. Recent empirical analysis shows that several vision–language models exhibit a pronounced bias toward textual inputs, often treating vision as auxiliary rather than co-equal evidence (Deng et al., 2025). This imbalance weakens multimodal grounding and increases susceptibility to attacks that exploit underweighted modalities or bypass cross-modal consistency checks.

### 4.2.4 Asymmetric Vulnerability Across Modalities

Multimodal language models can exhibit asymmetric susceptibility to adversarial perturbations across the modalities they process, where attacks on one modality are more effective or have a disproportionate influence on the model's final prediction compared to others. Such asymmetries can arise when modality-specific encoders or fusion mechanisms assign unequal weight or robustness to different input channels, causing certain modalities to dominate joint decision-making. Prior work on multimodal safety has shown that the visual modality can constitute a particularly brittle pathway for alignment, where adversarially crafted images can be leveraged to amplify harmful intent and induce safety violations even when text-only safeguards are in place (Li et al., 2025b). These findings highlight that modality-dependent robustness is uneven and that vulnerabilities in a single modality can disproportionately compromise multimodal reasoning. This asymmetry extends beyond static images: Hu et al. (2025b) show that the video modality introduces distinct

vulnerabilities, where distributing adversarial cues across frames enables jailbreaks that bypass defenses effective for single-image or text-only inputs.

### 4.2.5 Universal Representation Vulnerability

A recurring challenge in multimodal models is that their joint representations are expected to generalize across diverse tasks, prompts, and input distributions, which creates a shared attack surface. Empirical evidence shows that adversarial patterns can be optimized to generalize broadly across prompts and downstream tasks, remaining effective even when the attacker does not know the specific query or interaction context (Zhang et al., 2025d). Complementarily, Pandora's Box constructs a task-agnostic universal adversarial patch that transfers across multiple LVLMs and task settings, indicating systematic brittleness in shared multimodal representations across model instances (Liu et al., 2024a). As models increasingly rely on unified representational spaces for scalable multimodal reasoning, ensuring robustness across such varied scenarios remains difficult, leaving these shared representations vulnerable to reusable and transferable manipulations.

## 4.3 Modality-Specific Input Processing Vulnerabilities

Beyond cross-modal challenges, LLMs also exhibit vulnerabilities inherent in how they process and interpret data from individual modalities. These weaknesses often stem from the specific characteristics of each data type (e.g., high dimensionality of images, temporal nature of audio/video) or are inherited from the unimodal encoders (e.g., pre-trained vision or audio networks) used as building blocks. Attackers can exploit these modality-specific frailties even before information is fused, corrupting the input at its source.

### 4.3.1 Visual Input Processing

Multimodal language models exhibit significant vulnerabilities in their processing of visual inputs, arising from both the brittleness of vision encoders and their integration into language-driven reasoning pipelines. Small, often human-imperceptible perturbations to visual inputs can induce disproportionate changes in downstream MLLM outputs and behaviors, as adversarial signals injected at the vision-encoding stage propagate through multimodal fusion and language generation (Dong et al., 2023; Zhao et al., 2023). These vulnerabilities are frequently inherited from pre-trained vision backbones used as encoders, creating an attack surface that persists even when the language model itself is well-aligned. Beyond perception errors, recent work demonstrates that carefully crafted visual inputs can trigger unintended tool invocation or control behaviors in tool-augmented MLLMs, highlighting the security implications of visual processing failures (Fu et al., 2023). Other attacks exploit typographic overlays or visually rendered text that are misinterpreted by OCR or visual-text alignment components, effectively acting as covert visual prompts that manipulate downstream reasoning (Qraitem et al., 2024; Gong et al., 2025).

### 4.3.2 Audio Input Processing

The processing of speech, audio waveforms, and general sound events in multimodal language models introduces a distinct and increasingly exploited attack surface. Recent studies show that audio inputs are often less robustly protected than textual counterparts, allowing adversaries to induce harmful or unintended behaviors through carefully crafted speech or acoustic signals (Yang et al., 2024). In particular, vulnerabilities in speech recognition and audio–language alignment pipelines can be exploited via cross-lingual phonetics and pronunciation or accent variations that evade safety mechanisms while remaining intelligible to the model (Roh et al., 2025). Empirical red-teaming of audio multimodal models further reveals high attack success rates and inconsistent safety enforcement across acoustic inputs, underscoring that audio processing remains a comparatively weak link in current multimodal systems. While adversarial waveform attacks on acoustic models have been studied extensively in prior work Carlini & Wagner (2018); Zhang et al. (2017), such brittleness becomes especially consequential when these components are integrated into language-driven reasoning pipelines.

| Taxonomy Category | Primary Vulnerabilities Exploited | Representative Works |
|---|---|---|
| Signal Perturbations | Specific component weaknesses | Zhang et al. (2025d) |
| | Visual Input Processing; Embedding Space Weaknesses | Wang et al. (2024b) |
| Discrete Triggers | Visual Input Processing; Embedding Space Weaknesses; Universal Representation Vulnerability | Liu et al. (2024a) |
| | Visual Input Processing; Misalignment & Integration Failures | Cao et al. (2025) |
| Representation & Fusion Exploits | Embedding Space Weaknesses | Bagdasaryan et al. (2024) |
| | Misalignment & Integration Failures | Shayegani et al. (2023a) |

Table 2: Mapping Integrity Attacks to underlying vulnerabilities drivers

### 4.3.3 Textual Input Processing

Vulnerabilities in the interpretation of textual content remain critical in multimodal language models, particularly when text is processed indirectly through visual or multimodal pipelines. In such settings, models may over-attend to salient textual cues while failing to incorporate broader semantic context, leading to systematic misinterpretations. Recent work shows that typographic text rendered within images can be misprocessed by vision–language models, causing them to infer incorrect or unintended instructions despite the absence of explicit textual prompts (Qraitem et al., 2024; Gong et al., 2025). These failures highlight weaknesses in how textual information is extracted, normalized, and aligned with language reasoning components, especially when mediated by OCR or visual-text alignment modules.

### 4.3.4 Temporal Information Processing (Video/Sequential Data)

Multimodal language models face distinct vulnerabilities when processing temporally evolving inputs such as video, where maintaining coherent representations across sequential frames is essential for correct interpretation. Prior work indicates that disruptions to temporal consistency can propagate across time, causing errors introduced at a small number of frames to affect the model's understanding of an entire sequence. Such weaknesses are commonly associated with limitations in temporal aggregation and sequence-level reasoning, where models struggle to robustly integrate dynamic visual information over time (Huang et al., 2025a). Recent attacks explicitly exploit these limitations using flow-based methods that leverage optical flow to identify and perturb temporally salient regions, achieving high effectiveness by targeting a limited subset of frames rather than the full video stream (Li et al., 2024).

## 4.4 Instruction following & prompt-based vulnerabilities

A significant class of vulnerabilities in MLLMs stems directly from their core design objective: to understand and follow human instructions. Attackers exploit this inherent helpfulness by manipulating input prompts that are delivered via text, images, audio, or combinations thereof to elicit unintended, harmful, or restricted behaviors. These attacks often aim to subvert the model's safety alignments or hijack its generative capabilities for malicious purposes.

### 4.4.1 Exploiting Instruction-Following Nature

The same instruction-following capability that enables multimodal language models to perform complex tasks also introduces a critical vulnerability when adversarial or deceptive commands are introduced. Prior work shows that models can be manipulated by embedding malicious instructions that compete with or override user intent, causing the system to prioritize attacker-supplied guidance (Kimura et al., 2024). While

| Taxonomy Category | Primary Vulnerabilities Exploited | Representative Works |
|---|---|---|
| Unimodal Surfaces | Visual Input Processing; Jailbreaking & Safety Bypass | Qi et al. (2023) |
| | Audio Input Processing; Asymmetric Vulnerability Across Modalities | Yang et al. (2024) Roh et al. (2025) |
| Multimodal Composite Jailbreaks | Misalignment & Integration Failures; Embedding Space Weaknesses | Gong et al. (2025) |
| | Context Manipulation; Misalignment & Integration Failures | Wang et al. (2025d) |
| | Temporal Input Processing; Attention Mechanism Exploitation | Hu et al. (2025b) |
| | Jailbreaking & Safety Bypass; Misalignment & Integration Failures; | Li et al. (2025b) Wang et al. (2025b) |
| | Jailbreaking & Safety Bypass | Cheng et al. (2025) |
| Universal Jailbreak Triggers | Misalignment & integration failures; Visual input processing | Wang et al. (2024a) |
| | Embedding Space Weaknesses; Over Reliance on Specific Modalities | Geng et al. (2025) |

Table 3: Mapping Safety/Jailbreak Attacks to underlying vulnerability drivers

such instructions may appear explicitly in textual prompts, they can also be conveyed indirectly through non-textual modalities. In particular, attackers can encode instructions within images or audio inputs that are subsequently decoded and acted upon by the model as if they were legitimate textual commands, despite the absence of overt instruction text (Bagdasaryan et al., 2023; Gu et al., 2024). These attacks highlight that instruction-following behavior itself—rather than a specific modality—constitutes a fundamental vulnerability when models lack robust mechanisms for distinguishing intent, authority, and context.

### 4.4.2 Jailbreaking & Safety Bypass

Jailbreaking attacks target weaknesses in the safety and alignment mechanisms of large language models, seeking to induce outputs that violate intended content restrictions. Prior work shows that such failures often arise from the model's tendency to comply with cleverly framed instructions that exploit gaps or inconsistencies in safety training, for example through role-playing, hypothetical framing, or logical indirection. In multimodal settings, these vulnerabilities can be amplified when inputs from multiple modalities interact in ways that undermine unimodal safety checks (Liu et al., 2024c). Recent studies demonstrate that carefully constructed combinations of visual and textual inputs can jointly bypass safety mechanisms that would otherwise be effective in isolation, exposing weaknesses in cross-modal safety alignment (Liu et al., 2024d). These findings indicate that safety enforcement in multimodal models is not uniformly robust across modalities and can be circumvented through coordinated multimodal inputs.

### 4.4.3 Context Manipulation & Ambiguity Exploitation

Multimodal language models can be misled by adversarial manipulation of contextual information or by exploiting inherent ambiguities in language and cross-modal inputs. Prior work shows that by supplying misleading background context or framing information, attackers can steer the model's interpretation toward unintended conclusions or actions (Miao et al., 2025). Ambiguity can also be introduced deliberately through carefully constructed prompts or by combining multimodal inputs whose relationships are intentionally unclear or deceptive (Huang et al., 2025b). In such cases, models may default to an incorrect or attacker-favored interpretation, particularly when resolving conflicting or underspecified cues across modalities. Recent studies demonstrate that bi-modal adversarial prompts leveraging ambiguous visual–textual

| Taxonomy Category | Primary Vulnerabilities Exploited | Representative Works |
|---|---|---|
| Prompt Injection | Context Manipulation; Exploiting Instruction Following Nature; Visual + Textual Input Processing | Clusmann et al. (2025) |
| | Audio Input Processing; Exploiting Instruction-Following Nature; Context Manipulation | Hou et al. (2025) |
| System Instruction Manipulation | Exploiting Instruction Following Nature; Context Manipulation; Misalignment & Integration Failures | Wang et al. (2025a) |
| | Misalignment & Integration Failures; Over-Reliance on Specific Modalities | Zhang et al. (2025c) |
| Tool, Retrieval and Agentic Injection | Specific Component Weakness; Context Manipulation; Visual Input Processing | Fu et al. (2023) |
| | Context Manipulation; Exploiting Instruction-Following Nature; Misalignment & Integration Failures | Zhang et al. (2025a) |
| | Instruction Following Nature; Misalignment & Integration; | Gu et al. (2024) |

Table 4: Mapping Control & Injection Attacks to underlying vulnerability drivers.

relationships can effectively induce safety bypasses and unintended behaviors, highlighting weaknesses in contextual reasoning and disambiguation mechanisms (Ying et al., 2024).

#### 4.4.4 Attention Mechanism Exploitation

Attention mechanisms play a central role in how multimodal language models prioritize and integrate information across inputs, making them a critical attack surface (Vaswani et al., 2017). Prior work shows that adversarial inputs can be constructed to manipulate how attention is allocated, either by amplifying the influence of misleading features or by suppressing attention to semantically relevant cues. Such manipulation can occur within a single modality through self-attention, or across modalities by disrupting cross-modal attention alignment. Recent studies demonstrate that explicitly targeting attention dynamics can significantly alter model behavior, enabling attackers to steer outputs by biasing attention toward adversarially chosen elements (Wang et al., 2024c; Xie et al., 2024).

### 4.5 Architectural & Component-Level Vulnerabilities

Vulnerabilities in MLLM systems are not confined to input processing or instruction-following; they can also arise from internal architectural mechanisms and specific components within the model. Such weaknesses may stem from design choices in modules such as attention or fusion, or be inherited from pre-trained building blocks integrated into the system. Attacks that target these vulnerabilities aim to perturb internal representations and information routing, leading to systematic errors or unintended behaviors.

#### 4.5.1 Inherited Vulnerabilities from Pre-trained Components

Many multimodal language models rely on powerful unimodal encoders—such as vision, text, or audio backbones—that are pre-trained on large-scale datasets and subsequently integrated into larger multimodal systems. While such pre-training provides strong representational capabilities, it also introduces a pathway for adversarial weaknesses present in these source models to propagate into the full multimodal pipeline. Prior work shows that vulnerabilities in vision encoders, including susceptibility to adversarial perturbations, can

| Taxonomy Category | Primary Vulnerabilities Exploited | Representative Works |
|---|---|---|
| Data Poisoning | Inherited Vulnerabilities from pre-trained components; | Xu et al. (2024) |
| Backdoor Attacks | Data & Training Induced; Textual Input Processing | Yuan et al. (2025) |
| Fine-tuning, Adapter and Representation Poisoning | Inherited Vulnerabilites from pre-trained components; Specific Component Weaknesses; | Liu & Zhang (2025) |

Table 5: Mapping Poisoning & Backdoor Attacks to underlying vulnerability drivers

persist after integration and be exploited to induce incorrect or targeted behaviors in downstream multimodal models (Cui et al., 2024; Dong et al., 2023). As a result, attacks originally designed for standalone unimodal encoders may transfer to multimodal language models, exposing inherited weaknesses in the initial processing stages that are not mitigated by subsequent language reasoning components.

### 4.5.2 Specific Component Weaknesses

Beyond general architectural vulnerabilities, multimodal language models may exhibit weaknesses tied to specific components or auxiliary modules integrated into the system. In particular, lightweight adaptation mechanisms used for efficient fine-tuning—such as low-rank adapters—introduce additional attack surfaces that can be exploited independently of the base model. Prior work demonstrates that such adapters can be leveraged to implant backdoors with minimal computational overhead, enabling malicious behaviors to persist across downstream usage without modifying core model parameters (Liu et al., 2025a; Liang et al., 2025; Lyu et al., 2024). In addition, the choice of modality fusion components can influence robustness, as different fusion designs may vary in their susceptibility to adversarial manipulation or misalignment, potentially amplifying errors introduced at earlier processing stages.

### 4.6 Data & Training Induced Vulnerabilities

The security and reliability of MLLMs are strongly shaped by the data they are trained on and the training methodologies employed. Vulnerabilities may be introduced intentionally by adversaries through malicious manipulation of training or fine-tuning data, or arise unintentionally from biases, sensitive information, or spurious patterns embedded in large-scale multimodal corpora. Prior work demonstrates that attackers can inject a small number of poisoned samples containing specific trigger patterns paired with attacker-chosen outputs, causing models to learn latent correlations that remain dormant during standard evaluation yet activate reliably at inference time (Xu et al., 2024). In multimodal settings, such vulnerabilities are further amplified by cross-modal representation learning, where modality dominance and interaction effects can influence how poisoned behaviors are encoded and later activated (Han et al., 2024). These training-induced weaknesses are particularly difficult to detect and mitigate post-deployment, as the resulting malicious behaviors are deeply embedded in the model parameters and may only surface under out-of-distribution or trigger-specific conditions (Lyu et al., 2025).

## 5 Defense Mechanisms

This section presents an overview of defense mechanisms for MLLMs, and tying them according to the attack families in our taxonomy. In classical adversarial robustness, defenses largely focus on hardening modality encoders against bounded perturbations; in contrast, MLLM deployments introduce additional system-level failure modes in which untrusted multimodal content (e.g., OCR text in images, retrieved documents, or tool outputs) is inadvertently treated as higher-priority instructions. Consequently, defenses for MLLMs span multiple layers of the pipeline, including perception-layer robustness for integrity attacks, context and

instruction-handling mechanisms for jailbreak and injection attacks, and control-plane constraints for tool-using agents (C), with complementary safeguards for training-time compromise where applicable.

## 5.1 Input Preprocessing and Perception-Layer Hardening

A first line of defense operates at the input and perception layer by reducing sensitivity to low-level perturbations before or within modality encoders. Test-time transformations and feature-space compression can suppress small-magnitude perturbations that exploit weaknesses in visual or audio input processing, thereby mitigating integrity attacks based on continuous signal perturbations in our taxonomy. Feature squeezing is a representative approach that detects adversarial examples by comparing predictions before and after inexpensive input squeezes such as bit-depth reduction and smoothing (Xu et al., 2018). Such preprocessing-based defenses are lightweight and can be deployed without retraining, but their protection is typically partial under adaptive attackers and limited against attacks that do not rely on fragile pixel- or waveform-level noise. More recently, robustness of vision encoders in vision–language pipelines has been studied directly in the multimodal setting, for example by adversarially fine-tuning CLIP-style encoders to improve resistance against adversarial visual perturbations that propagate into downstream vision–language models (Schlarmann et al., 2024).

A more robust but costlier approach is adversarial training and robust optimization, which improves empirical robustness by training on worst-case (or proxy) adversarial examples. Foundational work on robust optimization and PGD-based adversarial training demonstrated substantial gains in resistance to bounded perturbations (Madry et al., 2019), and TRADES formalized the trade-off between natural accuracy and robustness via a principled objective (Zhang et al., 2019). Complementary approaches explore adapting prompt representations rather than model weights, showing that adversarial prompt tuning can improve robustness of vision language models to adversarially perturbed visual inputs without modifying the underlying encoders (Zhang et al., 2024). In MLLMs, these approaches most directly strengthen modality encoders and thus primarily mitigate integrity attacks rooted in perceptual signal manipulation, with partial benefits for representation-level vulnerabilities when the exploited failure originates in encoder embeddings rather than in higher-level instruction handling or fusion logic.

## 5.2 Certified Robustness for Encoders

Beyond empirical robustness, certified defenses aim to provide formal guarantees that model predictions are invariant within a specified perturbation set. Randomized smoothing offers probabilistic robustness certificates under $\ell_2$ perturbations and has been shown to scale to large neural models (Cohen et al., 2019). Recent work has begun extending certification techniques to vision language models, including incremental randomized smoothing methods that certify robustness of multimodal encoders under bounded perturbations (Nirala et al., 2024), as well as prompt-level certification strategies for medical vision language models (Hussein et al., 2024). Such certification most naturally applies to individual perception components within MLLM pipelines and therefore targets integrity attacks arising from bounded input perturbations. However, end-to-end certification of multimodal fusion, autoregressive decoding, and instruction-following behavior remains challenging, leaving representation-level exploits and higher-level safety or control attacks largely outside the scope of current guarantees.

## 5.3 Multimodal Input Validation and Specification-Based Gating

A distinctly MLLM-oriented defense direction is to validate multimodal inputs against explicit, application-provided specifications before they are allowed to influence the model's reasoning. Sharma et al. (2024) propose a defense specifically for image-based prompt attacks against MLLM chatbots, using a two-stage pipeline that (i) validates whether an input image conforms to expected constraints and (ii) performs prompt-injection defense to block malicious intent encoded in the image. This class of defenses directly targets safety bypasses and prompt injection attacks when adversarial instructions are introduced via the image channel (e.g., OCR-readable directives), and it can also reduce exposure to fusion-level vulnerabilities by preventing untrusted multimodal artifacts from reaching cross-modal reasoning in the first place. Beyond specification-based filtering, recent defenses also exploit cross-modal interactions themselves to disrupt jailbreak attempts,

for example by introducing adversarial visual perturbations that proactively interfere with malicious instructions before they influence the language model (Li et al., 2025a).

## 5.4 Instruction–Data Separation for Multimodal Contexts

A central vulnerability behind many injection attacks—amplified in multimodal settings—is the mixing of trusted instructions with untrusted context such as OCR outputs, retrieved documents, captions, or tool results. Defenses in this category enforce structured separation so that untrusted content is treated as data rather than executable directives. Liu et al. (2025b) formalize prompt injection attacks and defenses for LLM-integrated applications and provide a benchmarked evaluation framework that clarifies why naïve concatenation enables injected tasks to override target tasks. Building on this separation principle, Chen et al. (2024) propose structured queries that explicitly separate prompt and data channels and demonstrate improved robustness against prompt injection. Related defenses further refine instruction-data separation by shaping model preferences or introducing lightweight defensive tokens, demonstrating improved resistance to injected instructions under mixed trusted and untrusted contexts (Chen et al., 2025b;a). In our taxonomy, these defenses map most directly to control and instruction-hijacking attacks as well as many safety bypass scenarios, and they partially mitigate representation- and fusion-level exploits when the exploit relies on cross-modal context being reinterpreted as commands.

## 5.5 Detection-Based Defenses for Injection and Hijacking

Detection-based defenses aim to identify injected instructions or malicious intent embedded in untrusted multimodal context, including OCR-extracted text from images. In practice, detection and rejection can complement strict separation because not all applications can constrain inputs to a narrow specification, and attackers may embed prompt-like directives in visually plausible content. The two-stage approach of Sharma et al. (2024) explicitly includes prompt-injection defense to detect unsafe intent that bypasses initial validation, while the benchmarking methodology of Liu et al. (2025b) emphasizes evaluating defenses under diverse injection strategies and attacker capabilities. These defenses primarily mitigate instruction-hijacking and safety bypass attacks that exploit instruction-data confusability, but they may be bypassed by adaptive attacks that obfuscate or distribute malicious directives across modalities. Recent work has proposed deployable detection mechanisms aimed at identifying prompt injection attempts at inference time, including lightweight classifiers designed for real-time use in practical systems (Jacob et al., 2025). At the same time, empirical analyses highlight fundamental limitations of LLM-based detection under adaptive adversaries, showing that attackers can often evade detectors through paraphrasing, indirection, or multi-step instruction dispersion (Choudhary et al., 2025).

## 5.6 Control-Plane Defenses for Tool-Using Multimodal Agents

When MLLMs are embedded in agentic systems with tools (browsers, shells, retrieval, APIs), the dominant risk shifts from unsafe text outputs to unsafe actions. Defenses therefore must constrain tool invocation, detect malicious environmental artifacts, and prevent agents from being steered into attacker-chosen action sequences. Ayzenshteyn et al. (2025) propose proactive defenses based on deception and instrumentation—planting deceptive strings, honeytokens, and traps to detect and derail autonomous agent behavior—demonstrating the utility of such mechanisms against agentic attack workflows. These defenses map most directly to control-plane attacks in which adversarial information propagates into tool execution, including cases where multimodal inputs (screenshots, documents, OCR) serve as the carrier for adversarial instructions that would otherwise lead to tool misuse. More recent work extends control-plane defenses to end-to-end agent systems, proposing real-time monitoring and intervention mechanisms for computer-use agents (Hu et al., 2025a), as well as benchmark-driven evaluations that formalize attack and defense dynamics in LLM-based agents (Zhang et al., 2025b).

### 5.7 Poisoning/Backdoor Defenses

Data poisoning and backdoor threats motivate defenses that operate both during data curation and post-training verification. A common line of work aims to detect and remove poisoned samples using representation outliers, e.g., spectral signature methods that identify anomalous directions in learned feature space and filter suspicious training points (Tran et al., 2018). Complementary approaches attempt post-hoc trigger discovery and mitigation by reverse-engineering candidate triggers and identifying backdoored classes; Neural Cleanse proposes an optimization-based procedure to reconstruct potential triggers and then repair the model accordingly (Wang et al., 2019). At inference time, runtime input filtering can flag triggered behavior by measuring prediction consistency under strong perturbations, as in STRIP (Gao et al., 2019). Finally, model repair methods can reduce backdoor capacity by pruning neurons that are dormant on clean inputs followed by fine-tuning; Fine-Pruning demonstrates that combining pruning with fine-tuning can substantially weaken backdoors while largely preserving clean accuracy (Liu et al., 2018).

## 6 Limitations

This survey focuses on adversarial attacks against MLLMs, emphasizing attack objectives, mechanisms, and threat models. As a result, several limitations should be noted. First, our analysis is intentionally attack-centric. While we provide a structured overview of defense mechanisms in Section 5 to contextualize these threats, we do not claim to provide an exhaustive catalog of all possible mitigation strategies. Second, regarding scope, we prioritize peer-reviewed research that evaluates attacks on MLLMs with a language-model-based reasoning core. Adversarial studies targeting unimodal models alone or those aiming at content bias, fairness, and privacy leakage—are excluded unless explicitly framed as adversarial attacks on multi-modal systems. Third, the survey reflects the inherent skew in the current literature towards MLLMs. While we discuss Audio- and Video-based attacks where available, these modalities are currently less represented in the broader research landscape compared to image-text pairs. Finally, the taxonomy presented emphasizes dominant attack objectives. Real-world attacks may combine multiple mechanisms and exploit several vulnerabilities simultaneously, and thus may span multiple categories.

## 7 Broader Impact

This survey systematically catalogs adversarial attacks and vulnerabilities affecting MLLMs. While such taxonomies can lower the barrier to understanding and potentially reproducing known attacks, our intent is to support the development of more robust, secure, and trustworthy multimodal AI systems. By organizing attacks according to objectives, mechanisms, and threat models, we aim to help practitioners, system designers, and researchers anticipate realistic threats and reason about their root causes. Understanding how and why attacks succeed is a prerequisite for developing effective defenses, auditing deployed systems, and informing responsible deployment decisions.

We acknowledge the inherent tension between disseminating attack knowledge and the risk of misuse. Consistent with established practice in security research, we rely on peer-reviewed sources and emphasize analysis rather than operational guidance, avoiding step-by-step instructions or exploit-ready artifacts. Moreover, we include a dedicated discussion of defense mechanisms to contextualize attacks and highlight mitigation strategies. Finally, the real-world safety implications of multimodal attacks—such as misinformation, misuse of autonomous agents, and harm arising from model manipulation—underscore the urgency of this research area. We view this survey as a contribution toward responsible disclosure and proactive risk mitigation, rather than enabling malicious use.

## 8 Conclusions

This survey presented a systematic review of adversarial attacks on multimodal large language models, organized through a goal-driven taxonomy and complemented by a vulnerability-centric analysis. By classifying attacks according to their primary adversarial objective and considering different attacker knowledge assumptions, we provide a structured framework for organizing a rapidly growing and diverse body of literature.

The taxonomy is intended to offer a unifying perspective that cuts across modalities, attack implementations, and deployment settings. Across the surveyed literature, a substantial portion of existing attacks have been evaluated in vision–language model settings, reflecting the prevalence of image–text interfaces and the relative maturity of vision encoders in current multimodal systems. At the same time, attacks targeting audio–language and video–language models appear less frequently in the literature. We interpret this observation as an indication of uneven research coverage rather than a definitive assessment of comparative risk across modalities.

Our vulnerability analysis highlights several recurring weaknesses that are exploited across different attack families, including cross-modal misalignment, embedding-space fragility, modality-specific processing limitations, instruction-following ambiguities, and training-time data integrity issues. The surveyed works suggest that successful attacks often leverage combinations of these vulnerabilities rather than a single isolated weakness. In addition, attack objectives in multimodal systems frequently overlap: integrity failures may enable safety bypasses, control-oriented attacks may manifest as perception errors, and training-time poisoning can surface as inference-time integrity or control violations. Our taxonomy captures this structure by assigning each attack a primary objective while acknowledging secondary effects. Finally, the survey surfaces several open challenges suggested by patterns in existing work. Evaluation protocols for multimodal robustness remain fragmented, particularly outside vision–language settings. Reported defenses are often tailored to specific attack classes or threat models and may not generalize across modalities or deployment assumptions. We hope that the taxonomy, vulnerability mappings, and empirical summaries provided in this survey serve as a useful foundation for more systematic analysis and for the development of more robust and reliable multimodal language models.

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

## A  Survey Methodology

This section outlines the methodology used to identify, select, and analyze the literature for this survey, ensuring both its comprehensiveness and reliability.

### A.1  Search Strategy

To identify relevant publications, we conducted a systematic literature search using the Semantic Scholar database, selected for its broad coverage of computer science research. Our search employed key terms related to adversarial attacks and multimodal LLM's, including large language models and text, vision, audio, and

video modalities. To ensure relevance to current systems and threat models, we prioritized work published in recent years, while selectively incorporating earlier foundational studies when necessary for context. In addition, we performed targeted searches focusing on audio-based adversarial attacks to ensure adequate coverage of less-represented modalities.

### A.2 Inclusion and Exclusion Criteria

We include research works that satisfy all of the following conditions:

**Inclusion Criteria**

- **Target Model Scope:** The work studies attacks on MLLMs that:
  - process two or more input modalities (e.g., text, image, audio, video), and
  - incorporate a language-model-based reasoning or instruction-following core (e.g., GPT-4V, LLaVA, Gemini, Flamingo, Qwen-VL).

- **Attack Relevance to MLLMs:** The attack is evaluated on a multimodal LLM, even if the attack operates on a single modality (e.g., text-only, vision-only, or audio-only input surfaces). We refer to such attacks as unimodal attack surfaces on MLLMs and include them as baseline threats inherited by multimodal systems.

- **Adversarial Intent:** The work presents or analyzes attacks that intentionally induce one or more of the following:
  - integrity failures (incorrect, misleading, or targeted outputs),
  - safety or alignment violations (jailbreaks, policy bypass),
  - control or authority hijacking (prompt injection, tool misuse, agentic manipulation), or
  - persistent malicious behavior through training-time data poisoning or backdoors.

- **Attack Stage Coverage:** Both inference-time attacks (input, prompt, context, tool, or agent manipulation) and training-time attacks (data poisoning, backdoors, fine-tuning corruption) are included, provided they target multimodal LLM systems.

- **Scholarly Quality:** The work must be peer-reviewed and published in a recognized conference or journal venue.

**Exclusion Criteria**   A work is excluded if it meets any of the following conditions:

- **Unimodal Models:** Attacks targeting unimodal models only (e.g., text-only LLMs, vision-only classifiers, audio-only ASR systems) without evaluation on a multimodal LLM.

- **Non-LLM Multimodal Systems:** Works on multimodal systems without a language-model-based reasoning or instruction-following component, such as:
  - audio-visual classifiers,
  - detection or recognition pipelines, or
  - multimodal perception models without instruction following.

- **Non-Adversarial Failures:** Studies focusing solely on benign robustness, data bias, calibration, fairness, or generalization without an adversarial threat model, unless explicitly framed as adversarial attacks.

- **Non-Technical or Opinion Pieces:** Editorials, position papers without technical content, blog posts, or anecdotal reports.

- **Non-Peer-Reviewed Works:** Preprints, technical reports, or unpublished manuscripts that have not undergone peer review.

- **Non-English Language:** Only papers published in English are included.

### A.3 Summary of Adversarial Attacks in the Taxonomy on Multimodal Large Language Models

The following table consolidates all papers analyzed and categorized in the proposed taxonomy in Section 3.2.

Table 6: Overview of adversarial attacks on multimodal large language models

| Paper | Target Modalities | Attacker Knowledge | Impact on Performance |
|---|---|---|---|
| (Dong et al., 2023) | Image | Black-box | ASR: Bard 22%, Bing 26%, GPT-4V 45%, ERNIE 86% |
| (Qraitem et al., 2024) | Image + Text | Black-box | Accuracy drop up to 60% |
| (Liu et al., 2024a) | Image + Text | Black-box | Semantic similarity score up to 0.879 |
| (Cheng et al., 2024) | Image + Text | Black-box | Performance drop up to 42% |
| (Qraitem et al., 2025) | Image + Text | Black-box | ASR up to 100% (artifact ensembles) |
| (Cao et al., 2025) | Image + Text | Black-box | ASR: 44% (MCQ), 62% (open-ended) |
| (Xie et al., 2024) | Image | Black-box | Targeted ASR up to 98% |
| (Shayegani et al., 2023a) | Image + Text | Black-box | ASR: 85–87% (image triggers) |
| (Zhao et al., 2023) | Image + Text | Black-box | ASR: High Success Rate |
| (Yin et al., 2023) | Image + Text | Black-box | ASR up to 93.5% (task-dependent) |
| (Teng et al., 2025) | Image + Text | Black-box | ASR: 90% (open-source), 68% (closed-source) |
| (Yang et al., 2024) | Audio + Text | Black-box | ASR ~70% on harmful queries |
| (Roh et al., 2025) | Audio | Black-box | ASR increase up to +57% |
| (Li et al., 2025b) | Image + Text | Black-box | ASR: LLaVA 90%, Gemini 72% |
| (Miao et al., 2025) | Image + Text | Black-box | ASR: 85–91% across models |
| (Wang et al., 2025d) | Image + Text | Black-box | ASR up to 99% across benchmarks |
| (Yang et al., 2025) | Image + Text | Black-box | Average ASR 52%; ensemble 74% |
| (Jeong et al., 2025) | Image + Text | Black-box | Achieved upto 100% on LLaVA-1.5 13B |
| (Hou et al., 2025) | Audio + Text | Black-box | High Defense Success Rate (3%) |
| (Lee et al., 2025) | Image + Text | Black-box | ASR upto 90% |
| (Clusmann et al., 2025) | Image + Text | Black-box | ASR 67% (GPT4o) varies by model |
| (Zhang et al., 2025c) | Image + Text | Black-box | 86% |
| (Wang et al., 2025a) | Image + Text | Black-box | Increase of upto 30% |
| (Zhang et al., 2025a) | Image + Text | Black-box | Poison success rate of upto 92% |
| (Liao et al., 2025) | Image + Text | Black-box | Specific PII leakage upto 70% |
| (Tao et al., 2025) | Image + Text | Black-box | 83.5% for AntiGPT prompt |

| Paper | Target Modalities | Attacker Knowledge | Impact on Performance |
|---|---|---|---|
| (Lyu et al., 2025) | Image + Text | Black-box | Consistently high |
| (Liang et al., 2024) | Image + Text | Black-box | ASR >97% at 0.2% poisoning |
| (Hu et al., 2025b) | Video + Text | Black-box | Upto 96.5% LLaVA-Video-7B, depends on model |
| (Kimura et al., 2024) | Image + Text | Black-box | ASR up to 15.80% |
| (Wang et al., 2025b) | Image + Text | Black-box | ASR up to 94% |
| (Cheng et al., 2025) | Image + Text | Black-box | ASR up to 67.3% |
| (Gong et al., 2025) | Image + Text | Black-box | Average ASR 82.50% |
| (Liu et al., 2024c) | Image + Text | Black-box | ASR up to 72.14% |
| (Liu et al., 2024d) | Image + Text | Black-box | ASR up to 84.50% |
| (Huang et al., 2025a) | Video + Text | Black-box | ASR up to 55.48% (MSVD-QA), ASR up to 58.26% (MSRVTT-QA) |
| (Liang et al., 2024) | Image + Text | Black box | ASR > 97% |
| (Aichberger et al., 2025) | Image + Text | Gray-box | Universal Attacks 100% |
| (Liu & Zhang, 2025) | Image | Gray-box | ASR >99% |
| (Geng et al., 2025) | Image + Audio | Gray-box | ASR up to 86.6% |
| (Liang et al., 2025) | Image + Text | Gray-box | ASR up to 99.82% |
| (Zhang et al., 2017) | Audio | Gray-box | ASR almost 100% in ideal conditions |
| (Wang et al., 2024b) | Image + Text | Gray box | ASR up to 81.60% |
| (Xu et al., 2024) | Image + Text | Gray + Black-box | Poison ASR >95% (label flips) |
| (Xu et al., 2024) | Image + Text | Gray + Black-box | ASR > 95% |
| (Liang et al., 2025) | Text + Image | Gray + Black box | ASR up to 99.82% |
| (Wang et al., 2025c) | Image | White-box | Jailbreak ASR >88%; hallucination >98% |
| (Ying et al., 2024) | Image + Text | White-box | Average ASR upto 68.17% for MiniGPT4 |
| (Yin et al., 2025) | Image + Text | White-box | ASR upto 100% |
| (Lyu et al., 2024) | Image + Text | White-box | Image captioning around 0.97 |
| (Yuan et al., 2025) | Image + Text | White-box | ASR up to 100% (captioning) |
| (Gu et al., 2024) | Image + Text | White-box | Near-100% infection ASR |
| (Qi et al., 2023) | Image + Text | White-box | Jailbreak ASR up to 91% |
| (Wang et al., 2024a) | Image + Text | White-box | ASR: 96% (MiniGPT-4) |
| (Bagdasaryan et al., 2023) | Image + Audio | White-box | Not provided |
| (Carlini & Wagner, 2018) | Audio | White box | ASR almost 100% |
| (Li et al., 2024) | Video | White box | Almost 60% garble rate |
| (Wang et al., 2024c) | Text | White box | ASR up to 70.60% |
| (Liu & Zhang, 2025) | Image + Text | White box | ASR up to 99% |
| (Cui et al., 2024) | Image | White box | Caption Retrieval Recall Drop 90% |

| Paper | Target Modalities | Attacker Knowledge | Impact on Performance |
|---|---|---|---|
| (Han et al., 2024) | Image + Text + Audio + Video | White box | ASR > 96% |
| (Liu et al., 2025a) | Text | White box | ASR up to 95.8% - 100% |
| (Fu et al., 2023) | Image + Text | White-box | 98% |
| (Huang et al., 2025b) | Image + Text | White + Black-box | ASR up to 98.5% (transfer) |
| (Huang et al., 2025b) | Image + Text | White + Black-box | White Box ASR: 82.04% Black Box ASR: Upto 98.5% |
| (Zhang et al., 2025d) | Image + Text | White + Black-box | Performance drops from 78% to 44.85% for InstructBLIP Model |
| (Hao et al., 2024) | Image + Text | White + Black-box | ASR up to 77.75% (MiniGPT-4) |
| (Bailey et al., 2024) | Image + Text | White + Black-box | ASR > 80% |
| (Wu et al., 2025) | Image + Text | White + Black-box | ASR > 67% |
| (Bagdasaryan et al., 2024) | Image + Text + Audio + Thermal | White + Black + Gray box | ASR > 99.5% (Image-Bind/AudioCLIP) |

