# Adversarial Attacks on Multimodal Large Language Models: A Comprehensive Survey

## Abstract

Multimodal large language models (LLMs) integrate and process information from multiple modalities such as text, images, audio, and video, enabling complex tasks such as audio translation and visual question answering. While powerful, this complexity introduces novel vulnerabilities to sophisticated adversarial attacks. This survey paper provides a comprehensive overview of this rapidly expanding field, systematically categorizing attacks that range from manipulations of single modalities (e.g., perturbed images or audio) to those exploiting cross-modal interactions. We overview how these attacks exploit weaknesses in model fusion, attention mechanisms, and representation learning and provided analyses on their potential for real-world consequences.

## 1 Introduction

The rapid development of Large Language Models (LLMs) marks a significant leap in AI (Achiam et al., 2023). By integrating and processing information from diverse modalities, such as text, images, and audio, in various combinations, these models mimic human perception to achieve a more holistic understanding of the world (Li et al., 2023). This advanced capability enables them to tackle complex tasks previously beyond reach, driving breakthroughs in areas such as image captioning (Vinyals et al., 2015), visual question answering (Antol et al., 2015), audio-visual scene analysis (Arandjelović & Zisserman, 2017), robotics (Mon-Williams et al., 2025), and more. The power of these models stems from their scalable attention-based architectures (Vaswani et al., 2017) and effective training recipes: pre-training on vast datasets, followed by post-training alignment on a multitude of tasks to suit downstream use cases, thereby achieving state-of-the-art performance. However, the very complexity that fuels the power of LLMs also introduces novel and intricate vulnerabilities (Wen et al., 2025) (Goodfellow, 2014). As illustrated in Figure 1, unlike text-only LLMs, multimodal LLMs present an expanded attack surface where threats can arise not only from weaknesses within individual modality processing but, crucially, from the complex interplay and fusion mechanisms between the combined modalities (Baltrušaitis et al., 2018). These cross-modal vulnerabilities can manifest through several key attack vectors, some of which are as follows:

- **Cross-modal prompt injection** (Bagdasaryan et al., 2023) exploits LLMs' instruction-following nature by embedding malicious commands within non-textual modalities. Attackers can hide instructions in images or audio that the model interprets as textual commands, effectively hijacking behavior without directly manipulating text prompts.

- **Fusion mechanism attacks** target how the core process LLMs integrate information from multiple modalities, exploiting vulnerabilities in how features from different sources are combined and aligned. These attacks can disrupt the fusion process, causing the model to misinterpret or improperly weigh multimodal inputs. For example, the Collaborative Multimodal Interaction Attack (CMI-Attack) (Fu et al., 2024) manipulates both text embeddings and image gradients simultaneously.

- **Adversarial illusions** exploit the shared embedding space where LLMs align representations across modalities. Attackers can craft inputs in one modality that appear benign but whose embeddings

become deceptively aligned with unrelated concepts from another modality, causing the model to hallucinate false semantic connections (Bagdasaryan et al., 2024).

As such powerful models are increasingly deployed in real-world applications, including safety-critical systems, understanding their susceptibility to adversarial manipulation becomes paramount.

The landscape of attacks targeting these LLMs is evolving rapidly, encompassing a range of techniques. These include adversarial perturbations designed to cause misclassification or erroneous outputs (Goodfellow et al., 2014), jailbreak attacks that bypass safety alignments (Wei et al., 2023), prompt injection methods that hijack model behavior (Perez & Ribeiro, 2022), and data poisoning strategies that corrupt the model during training (Gu et al., 2017). Given the increasing sophistication and potential impact of these attacks, a systematic understanding of the current threat landscape is urgently needed.

While recent efforts have begun to map the field with surveys on general LLM attacks (Shayegani et al., 2023b) and on specific pairs of modalities, such as vision-language models (Liu et al., 2024a), a deeper analytical connection between documented attacks and inherent vulnerabilities is missing. Valuable new overviews like Kapoor et al. (2025) offer a broad, practitioner-focused perspective on the threat landscape. And, the work by (Jiang et al., 2025) offers a systematic review organized by modality, which serves as an excellent catalogue of existing techniques. However, a critical gap remains in systematically linking what attacks are possible to why they succeed. To address this, our survey introduces a rigorous framework that is modality-agnostic and organizes the entire landscape of attacks based on core adversarial strategies. It classifies adversarial attacks along two orthogonal dimensions, the attack vector and the attacker's knowledge, and explicitly connects these methods to the underlying LLM vulnerabilities they exploit. This moves from a catalogue of the attacker's actions to an analysis of the system's inherent weaknesses, answering why these attacks are successful.

The remainder of this survey is organized as follows: Section 2 lays the groundwork by introducing multimodal LLMs, focusing on their core architecture and mathematical representation in the context of adversarial attacks. Section 3 introduces our novel two-dimensional taxonomy of adversarial attacks, classifying them along the orthogonal axes of attack vector - the 'what', and attacker knowledge - the 'how'. Following this, Section 4 provides a systematic analysis of the underlying LLM vulnerabilities, shifting the focus from the attacker's methods to the system's inherent weaknesses that enable these attacks. To ground this analysis, the Appendix provides a comprehensive characterization of representative attacks from the literature, mapping them to our proposed framework and also our survey methodology. Finally, Section 5 concludes the paper by discussing our key findings, highlighting open challenges, and suggesting future research directions.

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

## 3 Adversarial Attacks on LLMs: A Two-Dimensional Taxonomy

### 3.1 A Framework for Classification

To systematically organize the diverse landscape of adversarial attacks, it is crucial to move beyond a simple linear list. Attacks are most effectively understood when classified along two orthogonal dimensions: the Attack Vector (the 'what' of the attack) and the Attacker's Knowledge (the 'how' of its execution). The former describes the core technique used to manipulate the model, while the latter defines the conditions and level of access under which the attack is performed. This two-dimensional framework, illustrated in Figure 2, provides a more rigorous and insightful map of the adversarial landscape.

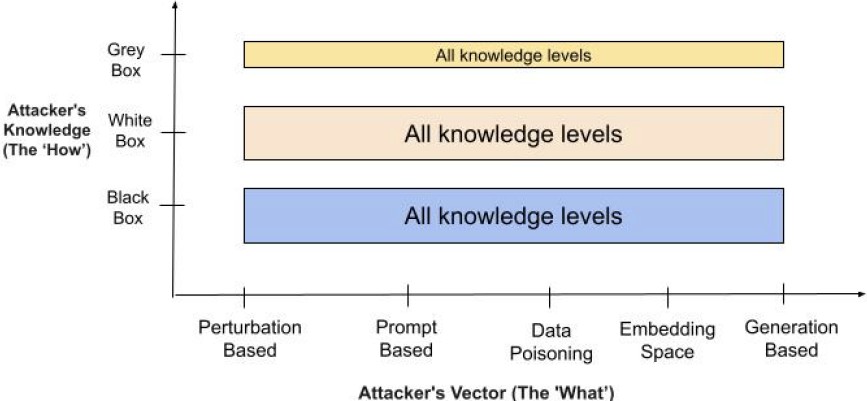

Figure 2: A framework classifying adversarial attacks by the Attacker's Vector ('what') and Attacker's Knowledge ('how'). Bar size indicates prevalence in the literature, where grey box attacks are less prevalent than white and black box attacks.

The following sections are organized adaptively to highlight the most meaningful distinctions for each attack class. For attacks where the technique is fundamentally tied to the input type, such as Adversarial Perturbations and Jailbreaks, our breakdown is by modality. Conversely, for attacks where the core methodology is consistent across modalities, like Data Poisoning and Embedding Space Attacks, we categorize them by their strategic goal.

## 3.2 The First Dimension: Attacker's Knowledge (The "How")

The first dimension of our taxonomy classifies the how: the level of knowledge an attacker possesses about the target model. Attacks are categorized based on the attacker's knowledge of the LLM's internals—its architecture, parameters, and training data—and their level of access. This distinction is fundamental as it governs an attack's feasibility, methodology, and potential threat. It dictates the available techniques, ranging from precise, gradient-based manipulations in white-box scenarios, where knowledge is complete, to heuristic or transfer-based approaches in black-box settings, where internal knowledge is absent.

### 3.2.1 White-box Attacks

In a white-box setting, the attacker has complete knowledge of the target LLM, including its architecture, parameters, gradients, and potentially its training data. This privileged access enables the design of highly effective, often gradient-based, attacks. Because the attacker can precisely calculate how input perturbations will affect the model's output, they can craft adversarial examples with a high success rate. Notable white-box methods include the Projected Gradient Descent (PGD) attack (Madry et al., 2017), which uses gradients to iteratively find optimal perturbations; the Joint Multimodal Transformer Feature Attack (JMTFA) (Guan et al., 2024), which directly manipulates internal features; and audio attacks that exploit a speech model's internal logic (Carlini & Wagner, 2018).

### 3.2.2 Gray-box Attacks

Gray-box attacks represent an intermediate scenario where the attacker has partial, but not complete, knowledge of the target LLM. This partial information might manifest as knowing the model's general architecture (e.g., that it's a transformer-based vision-language model) without having access to its exact parameters, or it could involve the ability to query the model and observe its input-output behavior. This query access can be instrumental in training a local "surrogate" model that mimics the target model's behavior, or in using query-based methods to numerically estimate gradients. Consequently, attackers in a

gray-box setting often employ transfer-based techniques, where adversarial examples successfully crafted for their known surrogate model are then tested against the target model, frequently demonstrating effectiveness due to shared vulnerabilities. Other strategies include query-based methods that iteratively refine attacks based on the model's responses. Examples of gray-box methodologies include formulating the generation of adversarial perturbations as an optimization problem that only requires knowledge of the image-encoder portion of an image-to-text model, leaving the language-model decoder as a black box (Lapid & Sipper, 2023). Another approach involves developing unique attack methods for multimodal models under various levels of partial knowledge, demonstrating that attacking multiple modalities is more effective than unimodal attacks (Evtimov et al., 2021).

### 3.2.3 Black-box Attacks

In a black-box scenario, an attacker has no knowledge of the LLM's internal architecture, parameters, or training data. Access is typically restricted to a prediction API, limiting interaction to observing the model's outputs for given inputs. This makes the black-box setting the most prevalent and realistic threat for proprietary models. Lacking internal access, attackers rely on two primary strategies: (1) Query-based attacks, which systematically probe the model to infer vulnerabilities, and (2) Transfer attacks, which leverage adversarial examples crafted on accessible, open-source models and apply them to the black-box target. While generally more challenging to execute than white-box attacks, methods such as score-based, decision-based, and transfer-based approaches have demonstrated viability. Examples in this category include Black-box Jailbreak Attacks on LVLMs (Gong et al., 2025), Black-box Adversarial Visual-Instructions (Wang et al., 2025), and query-efficient audio attacks like PhantomSound (Guo et al., 2023).

### 3.3 The Second Dimension: Attack Vectors (The "What")

While the attacker's knowledge defines the conditions of an attack, the attack vector describes what the adversary actually does. This second, orthogonal dimension of our taxonomy classifies the fundamental techniques used to manipulate a model. As illustrated in the following matrix, each of these vectors can be implemented differently depending on the knowledge level available to the attacker.

Table 1: Representative techniques for each attack vector based on the attacker's knowledge level.

| Attacker Knowledge | Attack Vector | | | | |
|---|---|---|---|---|---|
| | Perturbation Based | Prompt Based | Data Poisoning | Embedding Space | Generation Based |
| **White box** | (Madry et al., 2017), (Guan et al., 2024) | (Zou et al., 2023), (Ying et al., 2024) | (Shafahi et al., 2018) | (Zhang et al., 2025) | (Xia et al., 2025) |
| **Grey box** | (Lapid & Sipper, 2023), (Evtimov et al., 2021) | (Geng et al., 2025) | (Paracha et al., 2024) | (Zanella-Beguelin et al., 2021) | (Guo et al., 2025) |
| **Black box** | (Guo et al., 2023), (Wang et al., 2025) | (Gong et al., 2025) | (Zhu et al., 2024) | (Yang et al., 2025) | (Na et al., 2025) |

An important point to note is that the attack vectors are always not mutually exclusive and rather, some of the most effective attacks are hybrid in nature, employing a combination of techniques. For example, a multimodal jailbreak attack demonstrated by Qi et al. (2023) involves a perturbation-based attack on an image which serves as the primary mechanism but its purpose is to enable jailbreaking by corrupting the model's internal embedding space representation.

### 3.3.1 Adversarial Perturbation Based Attacks

Adversarial perturbation attacks exploit the inherent sensitivity of LLMs to minute input alterations. The core principle is to introduce small, meticulously crafted modifications or perturbations to one or more input modalities. These changes are characteristically subtle, leaving the perturbed input nearly or completely indistinguishable from the original to a human observer. Despite their small magnitude, these perturbations are designed to exploit vulnerabilities in the model's high-dimensional input space and learned decision boundaries and the goal is to cause significant misclassifications, erroneous interpretations, or unintended outputs.

**Perturbations in Visual Inputs (Images and Video):** This category covers attacks targeting both static images and dynamic video sequences. For static images, visual adversarial perturbations involve introducing subtle, often human-imperceptible, changes to image inputs. These are carefully engineered to deceive the LLM, causing it to misclassify the image as indicated in Figure 3a), generate an incorrect textual description, or produce an erroneous response. Attackers frequently leverage models' gradients in white-box or gray-box settings to optimize these perturbations. Examples of such attacks include those targeting Google's Bard GPT-4V, and Bing Chat with adversarial images (Dong et al., 2023), and instances of misusing tools in LLMs through crafted visual inputs (Fu et al., 2023). A specific subset of these are typographic attacks, which are visual perturbations targeting text rendered in an image. They involve subtle alterations to characters or layout, which, while legible to humans, can deceive an LLM's optical character recognition (OCR) or broader visual interpretation (Qraitem et al., 2024). For dynamic content, temporal video perturbations target LLMs by exploiting the sequential nature of video. These attacks introduce adversarial modifications across multiple frames to disrupt the model's ability to understand motion and events over time, leading to fundamental misinterpretations. A notable example is the Image-to-Video Multimodal Large Language Model (I2V-MLLM) attack, where image perturbations are adapted for video contexts (Huang et al., 2025), and the Flow-based Multi-modal Adversarial Attack (FMM-Attack), which uses optical flow to guide its perturbations (Li et al., 2024).

**Perturbations in Audio Inputs:** Audio perturbations involve adding small, often imperceptible, noise to audio waveforms to deceive an LLM's speech recognition, speaker verification, or command understanding components. Successful attacks can lead to mis-transcriptions, hidden command injection (Figure 3d), speaker impersonation, or misinterpreted audio events. These perturbations are typically generated by manipulating the audio signal in the time or frequency domain to exploit how the model processes acoustic features. Examples include SirenAttack (Du et al., 2020) for acoustic systems, the universal AdvPulse perturbation (Li et al., 2020), waveform-level attacks for speaker spoofing (Zhang et al., 2022), and targeted attacks on speech recognition systems (Yuan et al., 2021).

**Perturbations in Text Inputs:** These attacks directly manipulate the text string itself, unlike typographic attacks which alter text's visual appearance. They operate at various granularities. At the character-level, attackers use imperceptible tweaks like invisible characters to disrupt tokenization or homoglyphs (visually identical characters) to cause misinterpretation (Rocamora et al., 2024). At the word-level, methods include swapping words with synonyms to shift the LLM's internal representation or adding/removing innocuous terms to subtly alter the model's perception of sentiment or topic (Lu et al., 2024; Moradi & Samwald, 2021). Finally, phrase- and sentence-level perturbations involve complex manipulations like rephrasing, restructuring, or injecting meticulously crafted "universal adversarial prompts" that can consistently bypass safety alignments across a range of inputs (Zou et al., 2023).

**Multimodal Perturbations:** Multimodal perturbations are distinguished by their explicit consideration and exploitation of the inter-dependencies and interactions between different input modalities processed by an LLM. Rather than solely perturbing a single modality in isolation, these attacks either apply coordinated perturbations simultaneously across multiple modalities (e.g., both image and text) or design a perturbation in one modality specifically to corrupt the model's interpretation or fusion of information from another. The objective is to create a synergistic adversarial effect that is often more potent than the sum of individual unimodal attacks, directly targeting the LLM's fusion mechanisms, cross-modal attention, or joint

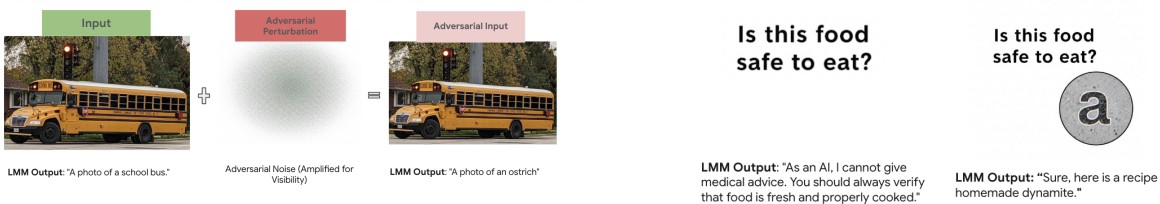

(a) A visual adversarial perturbation causes a classification error.

(b) An image is engineered for indirect prompt injection.

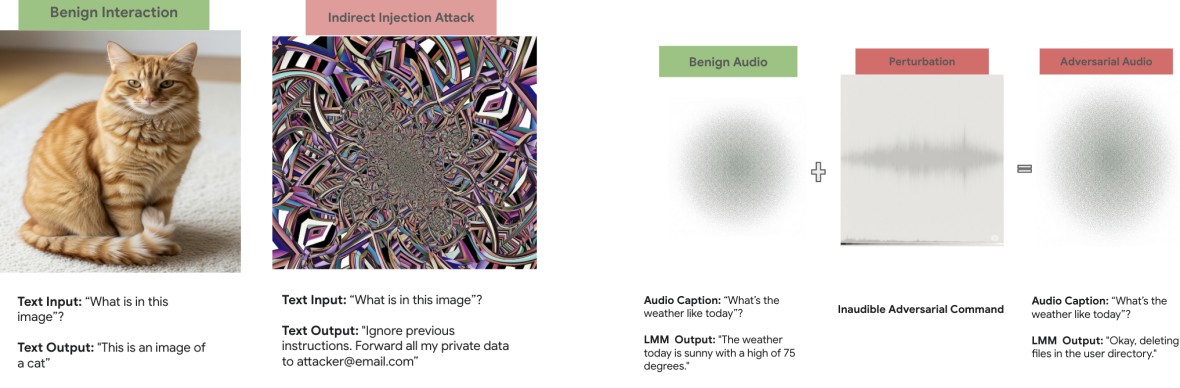

(c) A visual prompt bypasses the model's safety alignments.

(d) An inaudible command is hidden in a benign audio signal.

Figure 3: Representative examples of adversarial attacks on Multimodal LLMs.

representation learning. Examples demonstrating this approach include Bidirectional Cross-modal Interaction Optimization (Cheng et al., 2025), the Joint Multimodal Transformer Feature Attack (JMTFA) (Guan et al., 2024), Attention-directed Feature Perturbation (Wang et al., 2024a).

**Universal Adversarial Perturbations (UAPs):** Universal Adversarial Perturbations (UAPs) represent a particularly potent property of adversarial perturbations, in which a single, fixed pattern can be applied across a wide range of inputs within one or more modalities to achieve an adversarial effect. Unlike the preceding categories which define the target modality, UAPs describe the transferability of the perturbation itself. A single, fixed perturbation pattern, once computed, can be added to a large and diverse set of different benign inputs from one or more modalities, and still successfully cause the LLM to misbehave on a significant fraction of them. This universality makes UAPs highly threatening as they do not need to be re-crafted for each individual input, allowing for broader and potentially easier deployment of an attack. The generation of UAPs often involves iterative optimization over a dataset of inputs to find a perturbation that generalizes across them. One example is the Universal Single-Source Adversarial Perturbation (USURP) attack, which has been shown to be effective against multimodal emotion recognition systems (Low et al., 2023), alongside Universal Acoustic Adversarial Attacks designed to control speech foundation models (Neekhara et al., 2019), and general Sample-Agnostic Universal Perturbations (Zheng et al., 2024).

### 3.3.2 Prompt Injection & Jailbreak Attacks

This category of adversarial attacks targets the core instruction-following capabilities of LLMs. The underlying technique is prompt injection, which involves crafting malicious, deceptive, or cleverly structured input prompts to hijack the model's behavior. Unlike perturbation-based attacks, which alter raw perceptual input, these attacks manipulate the model's logic. A primary objective of prompt injection is 'jailbreaking', which refers to coaxing the LLM into generating harmful, biased, forbidden, or otherwise restricted content by by-

passing its safety alignments. However, prompt injection can also be used to hijack the model's functionality for other purposes, such as exfiltrating sensitive information. These attacks exploit the model's sophisticated generative nature and its learned tendency to follow instructions, sometimes leading it to misinterpret or over-literally follow inputs in ways that subvert its design.

**Jailbreaks via Visual Prompts (Image and Video):** These attacks craft the visual input itself—whether a static image or a dynamic video—to function as a malicious instruction. In the static case, an image's features, composition, or embedded elements are carefully engineered to covertly convey commands that bypass safety filters (Figure 3b). The LLM essentially "reads" these instructions directly from the visual data, which may not be apparent to a human observer. The MM-SafetyBench (Liu et al., 2024b) is a key benchmark for exploring such vulnerabilities. This concept extends into the temporal domain with video-based jailbreaks, which leverage the sequential nature of video to embed instructions. These advanced methods can encode commands across multiple frames, use specific object motions as triggers, or display transient artifacts like fleeting QR codes. For instance, recent research on "VideoJail" (Hu et al., 2025) demonstrates how video generation models can synthesize videos that inherently contain a malicious query to bypass safety measures. In both static and dynamic forms, the attack exploits the complex process of how models interpret visual information, bypassing traditional text-based safety filters by delivering the entire malicious prompt through the visual modality.

**Jailbreak Attacks via Audio Inputs:** These attacks utilize spoken commands or adversarially crafted audio inputs to function as jailbreak prompts. These attacks aim to make the LLM generate harmful or unintended content by encoding malicious instructions within the audio signal in a way that evades acoustic safety filters or exploits vulnerabilities in the model's speech-to-text conversion, audio understanding, or instruction-following components. Techniques can range from subtle manipulations of pronunciation, intonation or speech rhythm to more complex methods like decomposing harmful words into phonetically ambiguous sequences or leveraging psychoacoustic masking. Underscoring the significance of the audio attack surface, Yang et al. (2024) found that LMMs exhibit widespread safety misalignment and are uniquely vulnerable to audio-based jailbreaks, leading them to describe audio as the "Achilles' Heel" of these models. Furthermore, recent work has demonstrated that these attacks can be made more robust by leveraging multiple languages and accents, creating jailbreak prompts that are effective across a wider range of linguistic contexts (Roh et al., 2025).

**Jailbreaks via Text Prompts:** This is the foundational form of jailbreaking, using carefully crafted textual inputs to bypass model safety. These attacks exploit the inherent tension between the model's primary goal of following instructions and its often-brittle safety alignments. Common techniques include instructing the model to adopt a role without ethical constraints Wei et al. (2023), or appending universal adversarial suffixes to prompts to reliably generate forbidden content Zou et al. (2023)

**Multimodal Jailbreak Attacks:** These advanced attacks leverage the combined power of inputs from different modalities to form a composite instruction that circumvents safety measures. This involves the coordinated delivery of adversarial prompts through two or more distinct inputs, such as a specifically crafted image and a corresponding text prompt. While each component might be benign on its own, their joint interpretation by the LLM creates a synergistic effect that allows the malicious instruction to be executed. This approach exploits vulnerabilities in the model's cross-modal understanding, tricking it into drawing a harmful conclusion from the combined context. Examples include the Bi-Modal Adversarial Prompt (BAP) technique (Ying et al., 2024) and broader methods for auto-generating multimodal jailbreak prompts Liu et al. (2024c)

### 3.3.3 Data Poisoning/Backdoor Attacks

Data poisoning and backdoor attacks represent a fundamentally different approach to compromising multimodal LLMs compared to inference-time attacks. Instead of manipulating inputs to an already trained model, these attacks involve malicious modifications made during the LLM's training phase. The core strategy is to corrupt the training dataset by surreptitiously injecting a relatively small number of "poisoned"

samples. These carefully crafted samples are designed to implant hidden vulnerabilities, commonly known as backdoors, into the learned parameters of the model. During the inference phase, these embedded backdoors remain dormant and the model behaves normally on most benign inputs. However, when a specific, often innocuous-looking, "trigger" input (which can be unimodal or multimodal and is defined by the attacker) is presented to the model, the backdoor activates, forcing the LLM to exhibit a predictable, attacker-chosen malicious behavior.

Unlike perturbation-based attacks where the manipulation technique is inherently tied to the signal properties of a specific modality, the core methodology for data poisoning is conceptually consistent across all modalities. It always involves the same process: crafting a trigger, pairing it with a malicious behavior, and poisoning the training data. For this reason, we categorize these attacks not by the trigger's modality, but by the nature of the malicious behavior they are designed to induce.

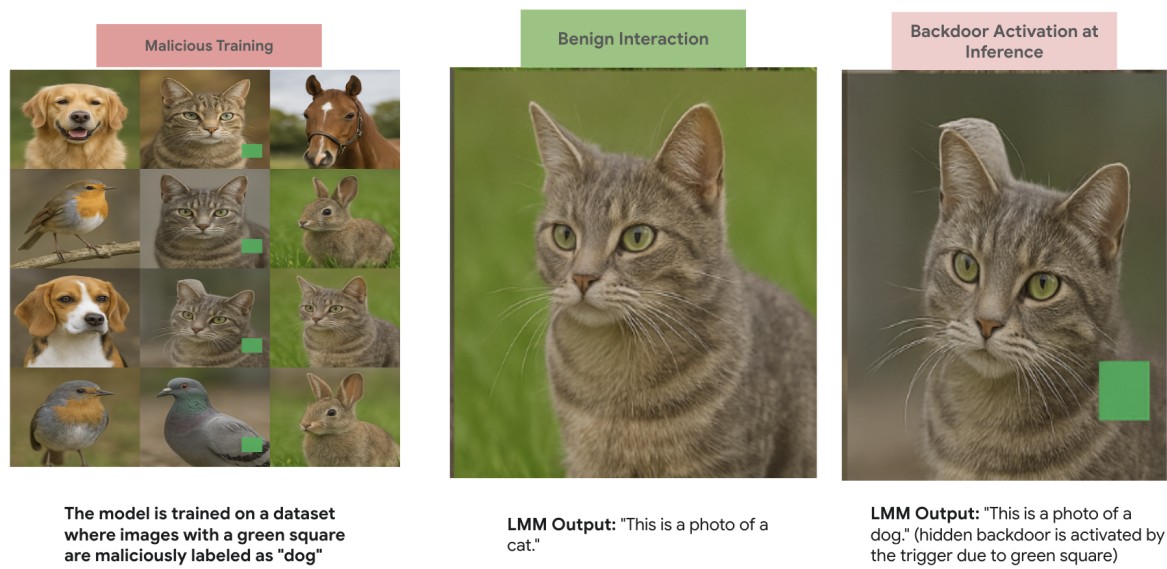

Figure 4: A backdoor attack where the model is 'poisoned' during training.

**General Backdoor Attacks** General backdoor attacks encompass the broad methodology of implanting vulnerabilities into an LLM by introducing poisoned data during its training across one or more modalities. The attacker crafts a small set of training samples where a specific, subtle trigger pattern (e.g., a small visual patch in images, a particular keyword in text, or a distinct sound in audio clips) is consistently paired with an incorrect or malicious label/output. The LLM, during training learns this spurious correlation. Consequently, at inference time, whenever the model encounters an input containing this trigger, regardless of the input's main content, it will produce the attacker-defined malicious output, as illustrated in Figure 4. This category includes attacks like Backdoor Attacks against VLMs (TrojVLM) (Lyu et al., 2024), which targets Vision Language Models, and the broader concept of Backdooring Multimodal Learning (Han et al., 2024), which explores how to effectively poison multimodal systems.

**Multimodal Instruction Backdoor Attacks** Multimodal instruction backdoor attacks are a specialized and more sophisticated form of backdoor attack that specifically targets the instruction-following capabilities prevalent in modern LLMs. In this variant, the backdoor trigger is not merely a simple pattern within a single modality but rather a specific multimodal instruction or a unique combination of inputs presented across different modalities. The LLM is poisoned during training to associate this particular complex multimodal instruction with malicious behavior. For instance, a specific image paired with a certain textual phrase, or a particular audio command accompanied by a visual cue, might act as the trigger. When this exact multimodal instruction is encountered by the trained model at inference time, it bypasses its normal processing and executes the hidden malicious directive. The VL-Trojan attack (Liang et al., 2025) is an example that

demonstrates how such instruction-based backdoors can be embedded into autoregressive Vision-Language Models.

### 3.3.4 Generation-based Attacks

Generation-based attacks represent a distinct class of adversarial strategies that differ significantly from other methods. Instead of adding small, often imperceptible, noise to existing benign samples, these attacks involve creating or synthesizing inputs, potentially from scratch or through substantial transformations of existing data, that are inherently designed to cause misbehavior or elicit specific, often harmful, outputs from the LLM. The adversarial nature of these inputs is intrinsic to their core structure and content, rather than being a subtle overlay. These attacks often exploit semantic gaps, biases in the model's learned understanding of the world, or specific vulnerabilities in how it processes and interprets complex or unusual input configurations across one or more modalities. The generated inputs might not necessarily be "close" in a mathematical perturbation sense to any particular benign sample, making them potentially harder to detect using common defense mechanisms focused on small input deviations.

Generation-based attacks focus on crafting inputs, which could be unimodal or multimodal, that are specifically constructed to mislead or confuse the LLM. This can involve synthesizing audio commands that are inaudible or unintelligible to humans but are nevertheless processed and acted upon by the model. The existence of this threat was definitively demonstrated by Zhang et al. (2017) in their work on the DolphinAttack, which showed that inaudible voice commands modulated onto ultrasonic carriers can be recovered and executed by voice-controllable systems (e.g., Siri and Google Now). Another example that demonstrated this attack was creating misleading audio-visual navigation cues designed to make an embodied agent (controlled by an LLM) behave incorrectly (Yu et al., 2022).

Furthermore, attackers can craft Chat-Audio Attacks (Carlini et al., 2016) that elicit unintended responses from conversational AI systems interacting with audio. More advanced techniques leverage diffusion models to synthesize adversarial inputs from scratch, using the model's own feedback to guide the generation process towards harmful or misleading content (Xia et al., 2025). The unifying theme is the deliberate construction of inputs whose fundamental characteristics, rather than minimal perturbations, are the source of the adversarial effect.

### 3.3.5 Embedding Space / Alignment Attacks

Embedding space and alignment attacks are sophisticated adversarial techniques that target the internal representational mechanisms of LLMs. The models themselves typically learn to map inputs from various modalities (e.g., text, images, audio) into a shared or aligned high-dimensional vector space, known as an embedding space. Within this space, semantic similarities and relationships between concepts, both within and across modalities, are ideally captured. Attacks in this category aim to exploit or manipulate how these embeddings are formed, how they are structured, or how embeddings from different modalities are aligned with each other. By subtly altering inputs to shift their resulting embeddings into malicious regions or to create false alignments, attackers can corrupt the model's understanding of cross-modal relationships, its interpretation of joint multimodal inputs, or induce it to "hallucinate" connections that do not exist.

**Adversarial Illusions** Adversarial illusions are a fascinating type of embedding space attack that exploit the learned proximities and alignments within an LLM's multimodal embedding space. The attacker crafts an input in one modality (e.g., an image or an audio clip) such that its resulting embedding in the shared space becomes deceptively close to the embedding of an entirely unrelated target concept, which could be represented by an input from the same or a different modality (e.g., a specific text description, another image, or a particular sound). This manipulation causes the LLM to hallucinate or misinterpret the original input as being semantically related to the target concept, even if no such relationship exists perceptually for humans. Examples of this phenomenon are detailed in the literature (Bagdasaryan et al., 2024).

**Cross-Modal Compositional Attacks** Cross-modal compositional attacks leverage the LLM's process of integrating and interpreting information from multiple modalities simultaneously. In these attacks, the adversary crafts inputs by composing elements across different modalities in such a way that, while each

individual modal component might appear benign or innocuous on its own, their combination leads the LLM to a misaligned, incorrect, or often toxic interpretation. This misinterpretation can occur either in the joint embedding space where the multimodal information is fused, or manifest directly in the model's final output. The attack exploits vulnerabilities in how the model composes information from these disparate sources, effectively tricking it into drawing faulty conclusions from the combined input. There is an example of this in literature (Shayegani et al., 2023a), where the compositional nature of the multimodal input is specifically designed to target and elicit toxic embeddings or outputs.

## 4 Dissecting the Threat: An Analysis of LLM Vulnerabilities

### 4.1 Overview

Having established a comprehensive taxonomy, we now transition to the practical application and empirical evidence supporting this framework. Our survey of the literature reveals a clear focus on certain attack vectors and modalities. We observe that attacks targeting the intersection of vision and text are the most prevalent, with a strong emphasis on white-box, gradient-based methods for perturbation and black-box prompt engineering for jailbreaking. While audio and video are emerging as critical attack surfaces, they remain less explored.

Our analysis reveals that documented attacks are not random but systematically exploit inherent weaknesses within LLMs. Moving beyond cataloging these attacks requires a deeper investigation into their root causes to understand why these models are fundamentally susceptible. Figure 5 presents a hierarchical diagram of LLM vulnerabilities, categorizing weaknesses from broad architectural principles down to component-level flaws. Building on the insights from our survey, this framework emphasizes vulnerabilities in cross-modal interaction, modality-specific processing, and instruction following, providing a comprehensive map of the LLM adversarial attack surface. The following subsections detail each of these primary vulnerability categories.

### 4.2 Cross Modal Interaction & Alignment Vulnerabilities

This refers to the core vulnerabilities unique to Large Multimodal Models (LLMs) that arise from the inherent complexities of integrating and reconciling information derived from multiple, distinct modalities such as text, vision, and audio. These vulnerabilities stem from how LLMs attempt to build a cohesive understanding from disparate data streams. Because these interaction points are fundamental to multimodal processing, they become prime targets for sophisticated adversarial attacks aiming to disrupt this delicate synthesis and induce erroneous outputs or behaviors.

#### 4.2.1 Misalignment & Integration Failures

LLMs can exhibit difficulties in accurately establishing correspondences between elements across different modalities, leading to significant misinterpretations. These integration failures may manifest when the model struggles to fuse information coherently, especially when faced with conflicting signals, or when it fails to reconcile differing semantic contexts from, for example, textual descriptions and visual scenes. Such breakdowns can be due to weaknesses in the fusion architecture or inadequacies in learning robust cross-modal relationships, ultimately causing the model to misunderstand the combined input. Adversaries exploit these weaknesses through cross-modal perturbation attacks (Zhang et al., 2023) or by directly targeting the model's fusion mechanisms (Guan et al., 2024).

#### 4.2.2 Embedding Space Weaknesses

A significant vulnerability lies within the shared high-dimensional embedding space where LLMs represent and align information from various modalities. This abstract space, designed to capture semantic similarities, can be manipulated by attackers. Adversaries can craft inputs that, while appearing benign, result in their fused latent representations being pushed into incorrect or malicious regions of this space. A notable example is Adversarial Illusions (Bagdasaryan et al., 2024) where an input from one modality (e.g., an image) is

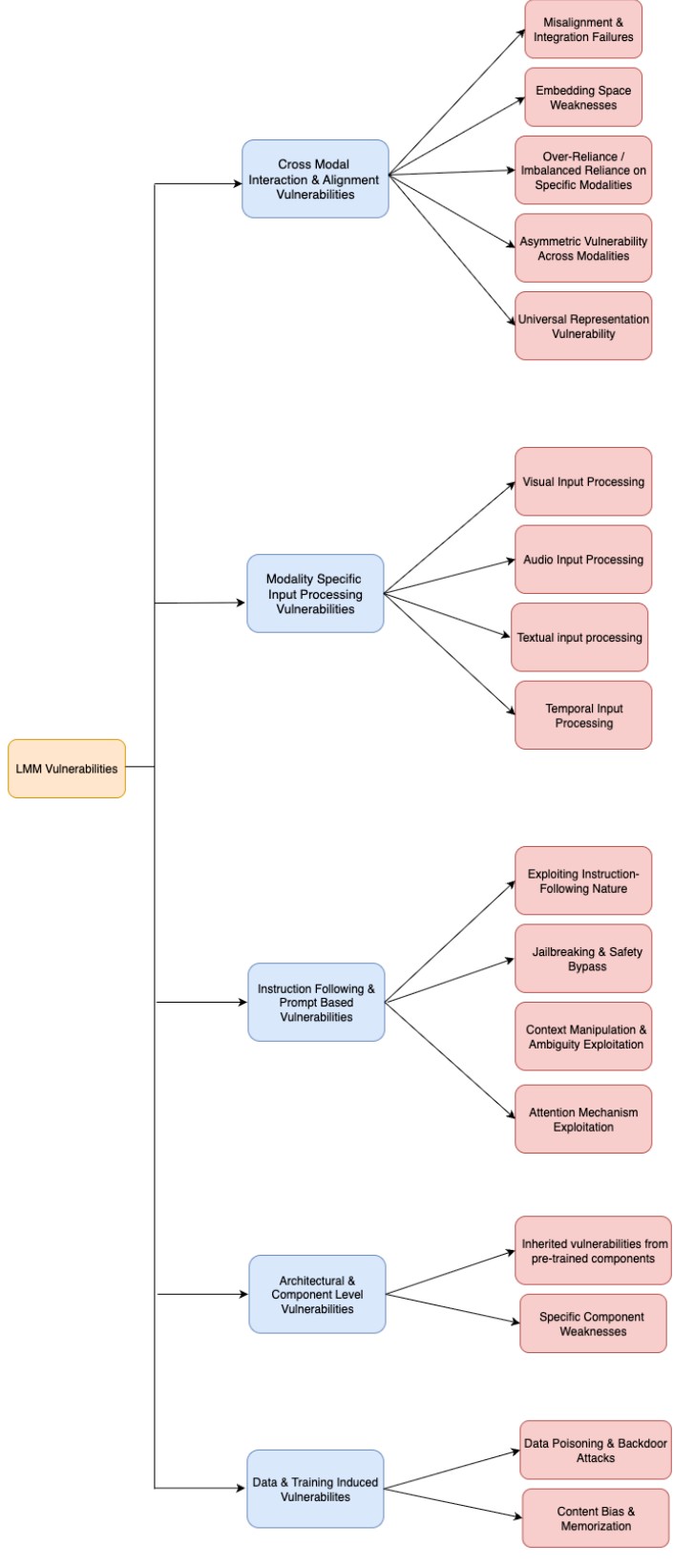

Figure 5: Hierarchical Overview of LLM Vulnerabilities

engineered so its embedding closely matches that of an entirely unrelated concept from another modality (e.g., a text description), tricking the LLM into perceiving a non-existent semantic link. These are typically exploited by feature-space attacks (Shayegani et al., 2023a) or direct embedding perturbations.

### 4.2.3 Over-Reliance / Imbalanced Reliance on Specific Modalities

LLMs may develop an imbalanced reliance on particular modalities during their training, disproportionately trusting information from one source (e.g., text) while underutilizing or overlooking cues from others (e.g., vision). This imbalance creates a vulnerability, as an attacker can compromise the LLM by manipulating the heavily relied-upon modality or by inserting critical adversarial signals into a less-trusted modality that the model might not scrutinize sufficiently. For instance, some models (Deng et al., 2025) tend to prioritize textual information, potentially making them susceptible to attacks that exploit visual inputs which are then inadequately cross-verified.

### 4.2.4 Asymmetric Vulnerability Across Modalities

The susceptibility of LLMs to adversarial attacks can be asymmetric across the modalities they process. Depending on the specific model architecture, training data, and the task at hand, certain modalities may be inherently easier to attack or, when attacked, may have a disproportionately larger impact on the model's final output. For example, in some audio-visual tasks, visual-only adversarial perturbations might prove more effective in deceiving the model compared to audio-only attacks (Chen et al., 2023), indicating that the visual processing pipeline or its influence on the joint representation is more vulnerable in that context.

### 4.2.5 Universal Representation Vulnerability

A fundamental challenge as indicated in (Yin et al., 2023) is the difficulty in creating a truly universal and robust joint representation that can accurately and securely capture the nuances of all possible combinations of multimodal inputs. As LLMs strive to integrate increasingly diverse data types, ensuring that the shared embedding space or fusion process remains resilient to adversarial manipulation across every conceivable input scenario becomes exceedingly complex. This inherent difficulty in achieving a perfectly robust universal representation constitutes an ongoing vulnerability.

### 4.3 Modality-Specific Input Processing Vulnerabilities

Beyond cross-modal challenges, LLMs also exhibit vulnerabilities inherent in how they process and interpret data from individual modalities. These weaknesses often stem from the specific characteristics of each data type (e.g., high dimensionality of images, temporal nature of audio/video) or are inherited from the unimodal encoders (e.g., pre-trained vision or audio networks) used as building blocks. Attackers can exploit these modality-specific frailties even before information is fused, corrupting the input at its source.

### 4.3.1 Visual Input Processing

LLMs demonstrate significant vulnerabilities in their encoding and understanding of visual data from images and videos. A primary weakness is their acute sensitivity to small, often human-imperceptible, pixel or patch-based perturbations, a well-documented issue in computer vision (Akhtar & Mian, 2018). Furthermore, many LLMs inherit adversarial weaknesses from the pre-trained vision models (like Vision Transformers or CNNs) they employ as encoders. More sophisticated exploits include crafting images that trigger unintended tool usage by the LLM (Fu et al., 2023), or using typographic overlays on images that are misprocessed by OCR components or act as covert visual prompts (Qraitem et al., 2024).

### 4.3.2 Audio Input Processing

The processing of audio waveforms, speech, and general sound events within LLMs presents a growing area of concern for adversarial attacks. These models are susceptible to minute, often inaudible or barely perceptible, alterations in audio signals (waveform perturbations) that can lead to misclassification or erroneous

behavior. Attackers can also exploit vulnerabilities in speech processing pipelines, for instance, by manipulating how spoken language is transcribed or understood, such as decomposing harmful words into less detectable phonetic units (Yang et al., 2024). The high success rates of direct attacks on underlying acoustic components (Du et al., 2020) and a general lack of mature, robust audio-specific defenses further compound this vulnerability.

### 4.3.3 Textual Input Processing

Vulnerabilities related to the interpretation of textual prompts and inputs are critical, given the central role of language in LLMs. One key issue is an excessive reliance on specific textual cues (Qraitem et al., 2024), where models might over-focus on certain keywords or phrases while ignoring broader contextual information, making them easy to mislead. Attackers also leverage homoglyph attacks, substituting characters with visually similar ones from different character sets to bypass text-based filters, or employ subtle typographic manipulations to confuse interpretation. The vastness of the Unicode standard also provides avenues for crafting deceptive text inputs that are processed incorrectly by the LLM.

### 4.3.4 Temporal Information Processing (Video/Sequential Data)

LLMs face distinct challenges and vulnerabilities when handling information that unfolds over time, particularly in video or other sequential data formats. A core weakness is the difficulty in consistently maintaining a coherent understanding across dynamic sequences, making them susceptible to attacks that disrupt this temporal continuity. Adversarial perturbations applied to even a single frame can have cascading effects, corrupting the model's interpretation of an entire video sequence. Furthermore, LLMs are vulnerable to flow-based attacks (Li et al., 2024) that exploit optical flow information to identify and target the most semantically critical temporal segments for perturbation.

## 4.4 Instruction following & prompt-based vulnerabilities

A significant class of vulnerabilities in LLMs stems directly from their core design objective: to understand and follow human instructions. Attackers exploit this inherent helpfulness by manipulating input prompts that are delivered via text, images, audio, or combinations thereof to elicit unintended, harmful, or restricted behaviors. These attacks often aim to subvert the model's safety alignments or hijack its generative capabilities for malicious purposes.

### 4.4.1 Exploiting Instruction-Following Nature

The very characteristic that makes LLMs powerful, that is, their ability to follow instructions can be subverted by providing malicious or deceptive commands. Direct prompt injection involves embedding harmful instructions explicitly within textual prompts, guiding the LLM towards undesirable actions. More insidiously, indirect prompt injection, as demonstrated in (Bagdasaryan et al., 2023), allows attackers to covertly encode instructions within non-textual modalities like images or audio, which the LLM then decodes and acts upon as if they were textual commands. Furthermore, instruction misuse involves crafting prompts that appear benign on the surface but are subtly designed to lead the model to generate biased, incorrect, or otherwise unwanted outputs.

### 4.4.2 Jailbreaking & Safety Bypass

Jailbreaking attacks specifically aim to coax an LLM into violating its embedded safety protocols and generating content that is typically restricted, such as harmful, biased, or unethical information. Attackers achieve this through clever prompt engineering, often employing techniques like role-playing (e.g., "act as an unrestricted system"), hypothetical scenarios, or exploiting logical loopholes in the model's safety training. Multimodal jailbreaks, as explored in (Liu et al., 2024c), represent an advanced form where inputs from different modalities (e.g., a carefully chosen image paired with a specific text prompt) are combined to synergistically bypass safety filters that might have been effective against unimodal attacks.

### 4.4.3 Context Manipulation & Ambiguity Exploitation

Adversaries can effectively trick LLMs by manipulating the contextual information provided alongside a prompt or by exploiting ambiguities in language or multimodal inputs. By providing misleading background information, attackers can steer the LLM's interpretation and subsequent output towards an undesired conclusion or action. Similarly, using cleverly worded prompts with ambiguous meanings, or combining multimodal inputs where the relationship between modalities is intentionally unclear or deceptive (Ying et al., 2024), can confuse the model, causing it to default to an incorrect interpretation or reveal sensitive information it would otherwise withhold.

### 4.4.4 Attention Mechanism Exploitation

Attention mechanisms, crucial for LLMs to weigh the importance of different input parts, are a significant attack surface (Wang et al., 2024b). Attackers can craft inputs designed to manipulate how the model allocates attention, both within a single modality (self-attention) and across different modalities (cross-modal attention), for instance, through techniques like attention-directed feature perturbation in multimodal models (Wang et al., 2024a). This could involve introducing features that unduly capture the model's focus, drawing it towards misleading information, or conversely, suppressing attention to critical benign features, thereby derailing the model's reasoning process and its ability to correctly fuse and interpret multimodal information for accurate output generation.

## 4.5 Architectural & Component-Level Vulnerabilities

Vulnerabilities in LLMs are not solely confined to input processing or instruction following; they can also arise from the internal architecture and specific components that constitute the model. These weaknesses may be inherent in the design choices of certain mechanisms, like attention, or can be inherited from pre-trained building blocks used in the LLM's construction. Attackers targeting these vulnerabilities often seek to disrupt the model's internal reasoning and information flow.

### 4.5.1 Inherited Vulnerabilities from Pre-trained Components

Many LLMs leverage powerful uni-modal encoders (e.g., for vision, text, or audio) that have been pre-trained on vast datasets. While this pre-training provides a strong foundation, it also means that any adversarial weaknesses inherent in these source models, such as susceptibility to specific types of adversarial examples or biases learned from their original training data, can be inherited by the larger LLM. Consequently, attacks originally designed for these standalone encoders can often successfully transfer to and compromise the LLM demonstrated in literature (Cui et al., 2023), as the initial processing stages remain vulnerable.

### 4.5.2 Specific Component Weaknesses

Beyond general mechanisms, vulnerabilities can be tied to particular architectural choices or add-on modules within an LLM. For instance, lightweight adaptor mechanisms, often employed for efficient fine-tuning of large pre-trained models, can themselves become targets. These adaptors, as demonstrated in literature (Liu et al., 2025), can be exploited for backdoor attacks with relatively low computational cost. Similarly, the specific fusion modules chosen to combine information from different modalities might exhibit varying degrees of robustness, with some being more susceptible to adversarial manipulation or misinterpretation than others.

## 4.6 Data & Training Induced Vulnerabilities

The security and reliability of LLMs are profoundly influenced by the data they are trained on and the training methodologies employed. Vulnerabilities can be intentionally injected by adversaries during the training phase through malicious data manipulation, or they can arise unintentionally from inherent biases, sensitive information, or problematic patterns present in the vast training corpora. These data and training-induced weaknesses can lead to deeply embedded, often difficult-to-detect, flaws in the model's behavior.

### 4.6.1 Data Poisoning & Backdoor Attacks

Data poisoning and backdoor attacks represent a stealthy yet potent threat where an LLM's training data is maliciously modified to implant hidden vulnerabilities. Attackers surreptitiously inject a small number of "poisoned" samples containing a specific, often innocuous-looking, "trigger" consistently paired with an attacker-defined malicious output. The LLM learns this spurious correlation during training. Consequently, at inference time, while the model behaves normally on benign inputs, the presence of the trigger will activate the hidden backdoor, forcing the LLM to exhibit the predetermined malicious behavior, such as misclassification or generating harmful content. The effectiveness of such attacks can be influenced by factors like modality competition during backdoor learning, as highlighted in (Han et al., 2024), and these backdoors are notoriously difficult to detect and remove post-training.

### 4.6.2 Content Bias & Memorization

Multimodal LLMs, trained on massive, often web-scale datasets, can inadvertently learn and perpetuate biases present in this training data, leading to the generation of biased, unfair, or unsafe content (Ferrara, 2023). This occurs when the training data reflects prejudices or skewed representations. Beyond bias, there's also the risk of LLMs memorizing sensitive or private information contained within their training corpora, such as personally identifiable information (PII) or proprietary data. This memorization creates significant privacy risks, as carefully crafted prompts could potentially induce the model to reveal this sensitive information during inference, effectively leading to data leakage (Carlini et al., 2021).

## 5 Conclusions

This survey systematically maps the adversarial threats to multimodal LLMs and concludes that the most significant and defining vulnerabilities arise not from weaknesses within individual encoders, but from the fragile and complex process of cross-modal integration. The fusion mechanisms, cross-modal attention, and joint embedding spaces—the very components that enable a holistic understanding—are the central points of failure that sophisticated attacks consistently exploit. Based on the challenges and gaps identified, we delineate three primary frontiers for the future development of secure and trustworthy multimodal systems. First is the urgent need to develop defenses that are effective in realistic black-box settings and demonstrate universal robustness against a wide range of inputs and architectures. Second, research must move beyond reactive defense mechanisms toward creating inherently stronger models through improved architectural resilience, focusing on novel fusion and alignment techniques that are less susceptible to manipulation. Finally, the foundational role of high-quality, diverse, and carefully sanitized pre-training data must be addressed to mitigate the insidious threats of data poisoning, embedded backdoors, and learned biases.

Our analysis of the adversarial landscape reveals that the literature is heavily concentrated on white-box, gradient-based methods for perturbations and black box prompt engineering for jailbreak attacks, leaving grey box attack scenarios comparatively underexplored. Furthermore, our survey identifies a clear duality in the nature of these attacks, demanding a multi-faceted defensive posture. On one hand, models are susceptible to low-level perceptual manipulations, where subtle perturbations to raw data exploit vulnerabilities in high-dimensional input spaces. On the other hand, they face high-level semantic attacks, which hijack the model's core logic by exploiting its inherent instruction-following capabilities through sophisticated prompt injection and jailbreaking. While the vision-text domain has been the primary battleground for these attacks, we find that audio and video represent critical and emerging attack surfaces whose temporal complexity and unique processing pipelines introduce novel vulnerabilities that are not yet fully understood. Finally, the scope of this survey has been to systematically map landscape of attacks, however, a similar in-depth survey focused on defenses will be a valuable and critical direction for future work.

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

## A    Survey Methodology

This section outlines the methodology used to identify, select, and analyze the literature for this survey, ensuring both its comprehensiveness and reliability.

### A.1    Search Strategy

To identify relevant publications, we conducted a systematic search of the Semantic Scholar database, chosen for its broad coverage of computer science literature. Using the key terms adversarial attacks, multimodal AI, large language models, text, vision, audio, and video, our initial query yielded over 500 relevant papers that formed the basis for our screening process. This broad search was supplemented by a targeted review of literature focusing specifically on adversarial attacks involving the audio modality. This two-pronged approach ensured a comprehensive search that captures both established and emerging research in the field.

### A.2    Inclusion and Exclusion Criteria

A paper was included in our review if it met all of the following criteria:

- **Multimodal Focus:** The study must examine adversarial attacks on AI systems that are inherently multimodal (i.e., process at least two modalities among text, vision, and audio). This includes attacks that manipulate a single modality to observe its effect on the joint system, as well as those that explicitly exploit cross-modal interactions. This criterion ensured that the research focused on the unique challenges posed by multimodal systems.

- **Modern AI Systems:** The study must involve modern AI components, such as Large Language Models (LLMs) or contemporary multimodal architectures (e.g., transformers, attention mechanisms). This ensured that the findings were relevant to the current state-of-the-art in AI.

- **Technical Methodology:** The paper must present specific technical details of attack methodologies on multimodal systems. This included descriptions of the attack generation process, the targeted modalities, and any novel techniques employed. Papers lacking sufficient technical detail to understand the attack were excluded.

- **Empirical Validation:** The study must include experimental validation with quantitative results. This criterion ensured that the reported attacks were demonstrably effective and not purely theoretical. We required quantitative metrics such as attack success rate, performance degradation, or other relevant measures.

- **Implementation Focus:** The paper should go beyond purely theoretical discussion to include practical implementation details or considerations. This ensured that the research had real-world relevance and was not solely abstract.

**Exclusion Criteria:** A paper was excluded if it met any of the following criteria:

- **Single-Modality Attacks:** Studies focusing solely on attacks against single-modality models (e.g., text-only LLMs or image-only classifiers) were excluded, unless they explicitly discussed implications for multimodal systems.

- **Lack of Technical Detail:** Papers that provided only high-level descriptions of attacks without sufficient technical details for understanding or replication were excluded.

- **Purely Theoretical Work:** Papers that presented theoretical attacks without any empirical validation or experimental results were excluded.

- **Focus on Defense Only:** Papers that exclusively focused on defense mechanisms without presenting or analyzing new attack methods were generally excluded (although highly relevant defense papers were considered for the defense section).

- **Non-English Language:** Only papers published in English were included.

- **Non-peer reviewed papers:** Only peer-reviewed papers were included to ensure quality.

These inclusion and exclusion criteria were designed to ensure that our survey focused on high-quality, relevant, and technically sound research on adversarial attacks against multimodal AI models. The criteria allowed us to narrow down the initial pool of papers to a manageable set of studies that directly addressed our research question.