# OpenReview forum: "Adversarial Attacks on Multimodal Large Language Models: A Comprehensive Survey"
_TMLR — Accepted by TMLR_

### Review · Reviewer_gXMn · 2025-11-17

**Summary Of Contributions:**

This paper presents a survey on adversarial attacks against multimodal large language models (MLLMs). It categorizes existing adversarial attacks along two dimensions: Attacker Knowledge and Attack Vectors (i.e., what the adversary actually does). The paper provides explanations and examples for each type of attack and appropriately analyzes the vulnerabilities of MLLMs from perspectives such as cross-modal interaction and modality-specific input processing.

There are some weaknesses, including:

1. The survey does not offer any insightful findings or in-depth discussions.

2. In terms of the number and diversity of papers reviewed, the coverage is insufficient. For each type of adversarial attack, only 12 representative works are selected for illustration, omitting a large body of relevant literature. Below are some recently published papers that were overlooked:
[1] Zhang et al., “Exploring the transferability of visual prompting for multimodal large language models.”
[2] Lu et al., “Set-level guidance attack: boosting adversarial transferability of vision-language pre-training models.”
[3] Wang et al., “Attention! Your Vision Language Model Could Be Maliciously Manipulated”
[4] Niu et al. “Jailbreaking attack against multimodal large language model”
[5] Jeong et al. “Playing the Fool: Jailbreaking LLMs and Multimodal LLMs with Out-of-Distribution Strategy”
[6] Zhao et al. “On evaluating adversarial robust- ness of large vision-language models”
[7] Wang et al. “White-box multimodal jailbreaks against large vision-language models”
[8] Sima et al. “VisCRA: A Visual Chain Reasoning Attack for Jailbreaking Multimodal Large Language Models”

3. Regarding the proposed classification dimensions:
(1) Dimension-1: for Attacker Knowledge, this is a common and generic classification method in the community. The survey merely adopts it without further organizing or analyzing the existing works from this perspective.
(2) Dimension-2: for Attack Vectors, while proposed as a novel classification dimension, its design is not convincing for the following reasons: (a) The goal of adversarial attacks is to cause model errors, and manipulating the embedding space is merely one technical means to achieve this, not an attack vector per se. Both generation-based and perturbation-based attacks can utilize embedding space as a technical approach. (b) The boundary between prompt-based and perturbation-based attacks is ambiguous. In Section 3.3.1, "Adversarial Perturbation Based Attacks," perturbation is broadly defined as modifications to inputs (image, audio, text, or video). This leads to significant overlap between the two categories. (c) Data poisoning is categorized separately. However, data poisoning is typically a training-phase attack, while adversarial attacks generally refer to inference-phase attacks. The two differ significantly in implementation. Additionally, backdoor attack is a specific form of data poisoning, but the paper appears to conflate these concepts in subsequent explanations.

**Additional Comments:**

N.A.

**Audience:**

No

**Audience Explanation:**

The number and scope of papers reviewed in this paper are insufficient, the classification criteria are unconvincing, and the proposed claims lack detailed substantiation. Therefore, I do not believe this survey would be particularly helpful to the TMLR audience. Refer to the detailed comments above.

**Broader Impact Concerns:**

No ethical concerns were identified.

**Claims And Evidence:**

No

**Claims Explanation:**

1. There is an inconsistency between the paper's title, content, and the "Inclusion and Exclusion Criteria." The title, "Adversarial Attacks on Multimodal Large Language Models: A Comprehensive Survey," defines the survey's scope as adversarial attacks on Multimodal Large Language Models (MLLMs). However, in the main text, when providing key examples of adversarial attacks, it cites papers [1, 2, 3] that are not related to Large Language Models. Furthermore, the provided "Inclusion and Exclusion Criteria" limit the selected articles to being "multimodal" but do not require them to be related to "large language models."
[1] Zhang et al. Cross-modal and cross-medium adversarial attack for audio.
[2] Chen et al. Push-pull: Characterizing the adversarial robustness for audio-visual active speaker detection.
[3] Du et al. Sirenattack: Generating adversarial audio for end-to-end acoustic systems.

2. The "Inclusion and Exclusion Criteria" state that "Only peer-reviewed papers were included to ensure quality," but the article in fact cites multiple non-peer-reviewed papers. For example:
[1] Zou et al. Universal and transferable adversarial attacks on aligned language models.
[2] Wang et al. Black-box adversarial attack on vision language models for autonomous driving.
[3] Guo et al. A grey-box attack against latent diffusion model-based image editing by posterior collapse.

3. The article does not provide a detailed argument, illustrative examples from papers, or comparative experimental data to support the claim that "the most significant and defining vulnerabilities arise not from weaknesses within individual encoders, but from the fragile and complex process of cross-modal integration."

**Requested Changes:**

1. Align scope and criteria: Strictly limit the survey to MLLMs and remove non-relevant or non-peer-reviewed papers.

2. Revise classification: Redefine the "Attack Vectors" dimension to resolve conceptual overlaps and inaccuracies.

3. Expand literature review: Include more and diverse works for each attack category.

4. Substantiate key claim: Provide concrete evidence for the central argument about cross-modal integration vulnerabilities.

---

### Review · Reviewer_q24q · 2025-12-23

**Summary Of Contributions:**

This survey paper provides a comprehensive overview of adversarial attacks on multimodal large language models (LLMs) that process text, images, audio, and video. The main contribution of this survey is a two-dimensional taxonomy that classifies attacks along two axes: (1) the attack vector (i.e., how is the attack performed, e.g. using perturbations, crafting prompts, etc.), and (2) the attacker's knowledge level (white, gray or black-box). The paper also presents a hierarchical framework (Figure 5) that categorizes weaknesses enabling these attacks, from architectural principles to component-level flaws. The survey covers attacks across multiple modalities and provides mathematical formalization of adversarial perturbations in the multimodal context in Section 2.2.

Strenghts:

- The proposed taxonomy offers a clear organization of the attack landscape, facilitating categorization across various model modalities and attack types. The formalization in Section 2.2 also defines the adversarial attack problem for multimodal systems, establishing a common notation that unifies discussion across different attack types. The survey methodology in Appendix A is also well-documented, with clear inclusion/exclusion criteria for related studies.

- The vulnerability analysis in Section 4 provides a useful shift in perspective from documenting attacks to understanding why multimodal systems are susceptible, with the hierarchical vulnerability diagram (Figure 5) providing a taxonomy of weaknesses (e.g., cross-modal interaction vs. architectural) that is often neglected in literature-focused surveys.

- The paper provides good coverage of emerging attack surfaces (audio, video) alongside the more established vision-text domain, acknowledging the asymmetric maturity of research across modalities.

Weaknesses:

- While the paper explicitly claims to provide "a comprehensive characterization of representative attacks from the literature" in the Appendix (page 2), the Appendix only contains the survey methodology.

- Section 4 analysis, while conceptually useful, remains at a high level and lacks quantitative evidence or case studies demonstrating how specific vulnerabilities are exploited by specific attacks.

- The discussion of defenses is minimal, relegated to a brief mention in the conclusions. Given the survey's goal of systematically understanding the threat landscape of adversarial, a dedicated section on defense mechanisms and their effectiveness would have been essential to ensure a comprehensive coverage of relevant literature.

**Audience:**

Yes

**Audience Explanation:**

This survey addresses a topic of significant and growing importance to the machine learning community. In particular, the deployment of multimodal LLMs in safety-critical applications (autonomous vehicles, healthcare, robotics) creates an urgent need for systematic understanding of their vulnerabilities. Relatedly, the proliferation of multimodal commercial systems like GPT-4o means that adversarial robustness in these domains is no longer a merely academic concern, and could benefit from the conceptual organization provided by the paper taxonomy. The identification of underexplored areas (gray-box attacks, audio/video modalities) is also a valuable contribution to benefit  future research in this area. Overall, these aspect make this work a potentially valuable reference for the community.

**Broader Impact Concerns:**

This survey paper raises some broader impact considerations that warrant attention. In particular, by systematically cataloguing attack methods and vulnerabilities, this survey could potentially lower the barrier for malicious actors to attack deployed multimodal systems. While the educational and defensive value of such surveys typically outweighs these risks, the authors should acknowledge this tension and emphasize responsible disclosure practices. Real-world safety implications of these attacks are not thoroughly discussed, despite being the main driver for the urgency of this research area. Finally, as already stated above, the near-total absence of defense discussion means practitioners receive no mitigation guidance. For these reasons, the paper would benefit from a dedicated Broader Impact Statement, acknowledging the value of understanding attacks for building robust systems and the responsibility that comes with publishing detailed attack taxonomies.

**Claims And Evidence:**

No

**Claims Explanation:**

While the paper does provides a modality-agnostic framework for understanding adversarial attacks on multimodal LLMs, some of its claims are not thoroughly supported.
In particular, the claim that the paper moves "from a catalogue of the attacker's actions to an analysis of the system's inherent weaknesses" is partially supported. While section 4's vulnerability analysis does provide an overview of why attacks succeed, categorizing various types of vulnerabilities, the connection between specific documented attacks and specific vulnerabilities could be made more explicit with concrete examples. Moreover, the claimed comprehensiveness is also questionable: the survey methodology indicates a screening of 500+ papers, but the final reference list contains approximately 80 citations, and the paper does not clearly state how many papers were ultimately included. Some recent notable works in the rapidly evolving field may be underrepresented as a result of this procedure.

**Requested Changes:**

Critical changes required for acceptance:
1. The paper promises in Section 1 that "the Appendix provides a comprehensive characterization of representative attacks from the literature, mapping them to our proposed framework." However, Appendix A only contains the survey methodology. The authors must either add the promised attack characterization or revise the claim in the introduction.
2. Better develop the connections between the attacks discussed in Section 3 and the vulnerabilities catalogued in Section 4. Currently, these sections are somewhat disconnected, with only passing mentions of which methods exploit which vulnerability. A more explicit discussion citing studies that empirically validate this connection would be necessary to provide convincing and clear evidence that the listed vulnerabilities are the primary drivers for the success of respective attacks.
3. Figure 2 should be significantly revised to accurately reflect the surveyed works. At the moment, bar colors and text "All knowledge levels" hold no meaning, and bar width (which is supposed to represent prevalence in the literature, as per the caption) does not vary across the X axis. A correct rendition of the graph should provide accurate numeric information about the prevalence of studies employing these methodologies for each Vector / Knowledge (X / Y) pair.
4. Add a dedicated section on defense mechanisms, even if brief. The current treatment (one sentence in conclusions) is insufficient for a "comprehensive survey." At minimum, categorize defenses analogously to attacks (e.g., input preprocessing, adversarial training, detection-based) and discuss their effectiveness against different attack vectors.

Additional changes that would strengthen the paper:

1. Include a quantitative meta-analysis or comparison of attack success rates where available. Table 1 lists representative techniques but provides no sense of their relative effectiveness, transferability, or practical threat level.
2. Expand coverage of audio-specific attacks and vulnerabilities with more technical depth. Section 3.3.1's audio perturbation discussion is notably shorter than the visual counterpart, and the "Achilles' Heel" characterization from Yang et al. (2024) deserves more exploration given its implications. If there is limited work in this area, say so explicitly.
3. The hierarchical vulnerability diagram (Figure 5) would benefit from citations to specific attacks that exploit each vulnerability node, making it a more actionable reference.

---

### Review · Reviewer_FnpH · 2026-01-07

**Summary Of Contributions:**

This paper discusses the topic of adversarial attacks on multimodal large language models.
The primary contribution of this paper is a comprehensive survey that focuses on multimodal settings rather than single-modality models.
The paper emphasizes analyzing why adversarial attacks succeed instead of only listing existing attack methods.
To support this goal, the paper introduces a modality-agnostic framework for organizing adversarial attacks.
This framework classifies attacks along two dimensions: attack vectors and the attacker’s knowledge.
The paper provides a clear architectural and mathematical description of multimodal LLMs and identifies potential attack surfaces.
The survey shows that many attacks exploit weaknesses in cross-modal fusion and attention mechanisms.
It further argues that vulnerabilities in shared embedding spaces are a major factor behind successful multimodal attacks.
The paper organizes these weaknesses into a hierarchical vulnerability taxonomy grounded in prior empirical studies.

**Audience:**

Yes

**Audience Explanation:**

If I understand correctly, this survey paper presents several new findings. For example, it argues that the primary vulnerability in multimodal large language models lies in cross-modal integration rather than in individual modality encoders. It also shows that diverse adversarial attacks can be systematically categorized within a two-dimensional framework defined by the attack vector and the attacker’s knowledge, providing a unified view of threat mechanisms. In addition, it notes that prior work has disproportionately focused on vision-text models, where white-box, gradient-based perturbations and black-box, prompt-based jailbreaks dominate, while audio, video, and gray-box settings remain relatively underexplored.

These findings could be valuable for researchers and engineers working on adversarial attacks against LLMs.

**Claims And Evidence:**

Yes

**Claims Explanation:**

The submission's main claims appear to be generally supported by accurate, clearly presented evidence.

The paper is appropriately positioned as a comprehensive survey, and it supports this positioning by describing its survey methodology by grounding the discussion in extensive prior work throughout the Introduction and Sections 2-4.

The proposed two-dimensional taxonomy is clearly defined in Section 3, and it is visually summarized through Figure 2 and instantiated with representative techniques in Table 1, after which it is applied consistently across the subsequent attack-vector discussions.
The claim that many attacks exploit vulnerabilities in cross-modal fusion, attention, and embeddings is supported by a structured vulnerability analysis  in Section 4 and representative examples from the literature.

However, some broader conclusions, such as the prevalence of certain modalities or whether cross-modal integration is the most critical vulnerability, rely mainly on qualitative synthesis rather than clear quantitative summaries.

Overall, the evidence appears to be convincing for a survey paper, but some claims would be stronger with clearer quantitative characterization of the surveyed literature.

**Requested Changes:**

* Clarify and lightly quantify prevalence claims
  The paper makes several claims about the prevalence of specific attack types, modalities, and threat models in the literature, such as the dominance of vision-text attacks and the emphasis on white-box perturbations over black-box jailbreaks. These trends appear plausible and are visually suggested by Figure 2, but the supporting evidence remains largely qualitative. The paper would be stronger if it included a brief quantitative snapshot of the surveyed literature, such as approximate counts or percentages by modality and attack vector, for example, in a table in Section 4.1.

* Better motivate the choice of the two-dimensional taxonomy
  The proposed two-dimensional taxonomy, organized by attack vector and attacker knowledge, is clear and easy to follow. However, the rationale for choosing these dimensions over alternative organizational schemes remains implicit. The paper would benefit from a short, explicit discussion of why these dimensions are particularly well suited to multimodal LLMs. For example, a brief comparison with modality-based or task-based taxonomies in Section 3 would help clarify the advantages of the proposed framework.

* Align the conclusion on cross-modal integration with the presented evidence
  The conclusion states that vulnerabilities in cross-modal integration represent the most critical weakness in multimodal LLMs. This claim is supported by the structured vulnerability analysis in Section 4, but the evidence is primarily qualitative. In my view, the paper could better align the evidence and the claim by refining the wording to emphasize consistency across surveyed studies. For example, framing cross-modal integration as one of the most frequently or systematically exploited vulnerabilities would better match the presented analysis.

* Make the connection between the taxonomy and representative attacks more explicit
  The appendix presents a useful catalog of representative attacks. However, the link between these attacks and the proposed taxonomy and vulnerability hierarchy remains implicit. The paper would be clearer if each representative attack were explicitly mapped to its attack vector, attacker knowledge level, and vulnerability category.


### ##Minor comments

* Clarify and strengthen the comparison with prior survey work
  The Introduction already positions the paper relative to several closely related surveys, but the comparison remains brief and would benefit from a clearer discussion of how the proposed two-dimensional taxonomy and vulnerability-focused perspective differ from, and complement, prior survey work. In addition, the following survey papers could be included, as they do not appear to be cited in the current version, if appropriate:

  - Mao+, "From LLMs to MLLMs to Agents: A Survey of Emerging Security and Jailbreak Issues" (2025).
  - Li+, "Security Concerns for Large Language Models: A Survey" (2025).
  - Yeo+, "Multimodal Prompt Injection Attacks: Risks and Defenses for Modern LLMs" (2025).
  - Wen+, "Investigating Vulnerabilities and Defenses against Audio-Visual Attacks with an Emphasis on Multimodal Models" (2025).　(cited but could be discussed more explicitly)

* Make the limitations of the survey more explicit and easier to locate
  The paper already acknowledges several limitations of its scope and methodology, including its focus on attack-side analysis and its emphasis on surveyed literature with empirical validation, as discussed in the Conclusion and the survey methodology appendix.
However, the paper would benefit from making them more explicit and easier to find, for example, by briefly consolidating them in a clearly marked section as Limitations.

* More explicitly connect the vulnerability analysis to implications for defense
  The paper focuses primarily on attacks, and its taxonomy and vulnerability analysis naturally suggest implications for defense. If I am not missing something, these implications are currently only briefly mentioned in the conclusion. The paper would be strengthened by a brief, structured discussion linking major vulnerability categories, such as fusion-level or embedding-level weaknesses, to corresponding defense directions.

---

### Decision · Action_Editor_YHH6 · 2026-02-27

**Recommendation:** Accept with minor revision

**Additional Comments:**

The authors already implemented many updates and the quality of the survey improved a lot. Nevertheless, they should make a minor update to the main text to improve accessibility:
(1) add a one-paragraph quantitative summary in the main text reporting the final number of included papers and simple counts by modality, knowledge level, and objective;
(2) briefly reiterate the paper inclusion policy in the main text (did I miss it?)

**Audience:**

Yes

**Audience Explanation:**

Yes. With adversarial risks in multimodal LLMs the survey addresses a timely and important topic, offering a structured taxonomy, vulnerability analysis, and an overview of defenses. Its modality-agnostic framing and identification of underexplored areas, is relevant for parts of TMLR's audience.

**Claims And Evidence:**

Yes

**Claims Explanation:**

Overall, the core claims are now largely supported and clearly presented. The revised taxonomy (organized by attacker objective and knowledge), the explicit attack-vulnerability mappings (Tables 2–5), the defense section, knowledge level, and reported ASR provide support for the main claims.

---

> ### Author Response · Authors · 2026-03-14
> **Camera Ready Submission and Addressing Feedback**
>
> We thank the reviewers and the Action Editor for their careful review and constructive feedback throughout the revision process. We have also addressed both points raised:
>
> 1. We have added a one-paragraph quantitative summary at the end of Section 3.1, reporting the total number of papers analyzed (88), along with breakdowns by primary adversarial objective, target modality, and attacker knowledge level across the 65 empirically characterized works consolidated in Table 6 (Appendix A.3)
>
> 2. We have also added a brief statement of the paper inclusion policy in the Introduction (end of Section 1), summarizing the scope and selection criteria with a reference to the full methodology in Appendix A. This ensures that readers encounter the inclusion policy early in the main text without needing to consult the appendix.
>
> We have also uploaded the camera-ready version of the paper incorporating these changes.